# Chronic pain induces generalized enhancement of aversion

Qiaosheng Zhang[1,2], Toby Manders[1,2], Ai Phuong Tong[1,2], Runtao Yang[1,2], Arpan Garg[1,2], Erik Martinez[1], Haocheng Zhou[1], Jahrane Dale[1,2], Abhinav Goyal[1,2], Louise Urien[1,2], Guang Yang[1], Zhe Chen[2,3], Jing Wang[1,2]*

[1]Department of Anesthesiology, Perioperative Care and Pain Medicine, New York University School of Medicine, New York, United States; [2]Department of Neuroscience and Physiology, New York University School of Medicine, New York, United States; [3]Department of Psychiatry, New York University School of Medicine, New York, United States

**Abstract** A hallmark feature of chronic pain is its ability to impact other sensory and affective experiences. It is notably associated with hypersensitivity at the site of tissue injury. It is less clear, however, if chronic pain can also induce a generalized site-nonspecific enhancement in the aversive response to nociceptive inputs. Here, we showed that chronic pain in one limb in rats increased the aversive response to acute pain stimuli in the opposite limb, as assessed by conditioned place aversion. Interestingly, neural activities in the anterior cingulate cortex (ACC) correlated with noxious intensities, and optogenetic modulation of ACC neurons showed bidirectional control of the aversive response to acute pain. Chronic pain, however, altered acute pain intensity representation in the ACC to increase the aversive response to noxious stimuli at anatomically unrelated sites. Thus, chronic pain can disrupt cortical circuitry to enhance the aversive experience in a generalized anatomically nonspecific manner.

*For correspondence: jing.wang2@nyumc.org

Competing interests: The authors declare that no competing interests exist.

## Introduction

Chronic pain exerts a profound influence over daily life by impacting a range of sensory and affective behaviors. It is associated with enhanced response to noxious stimuli, leading to symptoms of allodynia and hyperalgesia at the site of tissue or nerve injury (*Basbaum et al., 2009*; *Latremoliere and Woolf, 2009*). The mechanisms for such sensory and affective hypersensitivity at the site of chronic pain have been well investigated. However, conditions such as fibromyalgia and persistent postoperative pain raise the possibility that chronic pain may also increase the aversive reaction towards noxious stimuli in an anatomically nonspecific distribution (*Scudds et al., 1987*; *Petzke et al., 2003*; *Kehlet et al., 2006*; *Kudel et al., 2007*; *Scott et al., 2010*). If confirmed, this generalized form of enhancement in pain aversion, as opposed to anatomically specific hypersensitivity, can greatly expand our understanding of the impact of chronic pain on behavior.

The anterior cingulate cortex (ACC) has a crucial role in the affective-aversive experience of pain (*Lubar, 1964*; *Foltz and White, 1968*; *Turnbull, 1972*; *Talbot et al., 1995*; *Craig et al., 1996*; *Rainville et al., 1997*; *Koyama et al., 2000*; *Johansen et al., 2001*; *Koyama et al., 2001*; *LaGraize et al., 2006*; *Qu et al., 2011*). The ACC receives nociceptive inputs from the medial thalamus as well as from other cortical regions (*Vogt and Sikes, 2000*; *Shyu et al., 2010*). Individual ACC neurons can respond to noxious stimuli by increasing firing rates (*Sikes and Vogt, 1992*; *Yamamura et al., 1996*; *Hutchison et al., 1999*; *Kung et al., 2003*; *Iwata et al., 2005*; *Kuo and Yen, 2005*; *Zhang et al., 2011*) to provide evaluation for the intensity of acute pain (*Coghill et al., 1999*; *Büchel et al., 2002*). While previous studies have demonstrated that the ACC is necessary

and sufficient for the acquisition of stable aversive learning in the chronic pain condition (*Johansen et al., 2001*; *Qu et al., 2011*; *Barthas et al., 2015*; *Navratilova et al., 2015*), its role in the aversive response to transient acute pain signals is less well characterized. Furthermore, chronic pain has been shown to induce synaptic plasticity in ACC neurons, resulting in hypersensitivity at the site of injury through descending modulation (*Wu et al., 2005*; *Li et al., 2010*; *Koga et al., 2015*). It is unknown, however, if chronic pain can also impair ACC functions to alter the aversive response to noxious stimuli in an anatomically nonspecific manner. Using a multidisciplinary approach by combining optogenetics, in vivo electrophysiology and machine-learning decoding analysis, we found that chronic pain can disrupt acute pain representation in the ACC to induce generalized, anatomically nonspecific enhancement of aversion.

## Results

### Chronic pain can enhance the aversive response to noxious inputs at anatomically disparate sites

Conditioned place aversion (CPA) and conditioned place preference are well-established assays to test aversive learning as well as the negative reinforcement of analgesia in rodent chronic pain models (*Johansen et al., 2001*; *King et al., 2009*; *Navratilova et al., 2012*; *Daou et al., 2013*). These tests, however, have rarely been used to assess the aversive value presented by acute pain signals. Based on the concept of quantitative evaluation of aversion developed by standard CPA protocols (*Johansen et al., 2001*; *Johansen and Fields, 2004*), we constructed a brief 2-chamber CPA test for rats (*Figure 1A*). During the baseline (preconditioning) phase (10 min) of the test, rats were allowed free access to both chambers, and the time spent in either chamber was recorded as baseline. Next, during a brief conditioning period (10 min), we paired each chamber with a distinct laser stimulus directed at the hind paw of a freely-moving rat. The stimulus was classified as non-noxious (NS), low-intensity noxious (LS), or high-intensity noxious (HS). The intensity of the stimulus corresponded to the power output of the laser. Higher laser output transferred more heat to induce more intense thermal pain (*Figure 1—figure supplement 1*). At the behavioral level, NS did not typically elicit withdrawals within 5 s of stimulation (<5%), whereas HS and LS both elicited paw withdrawals 100% of the time. In addition, HS elicited withdrawals with a shorter latency than LS (*Figure 1B*). After this brief conditioning phase, we immediately tested the rat's aversive response by measuring the time spent in each of the two treatment chambers during a test (postconditioning) phase (10 min). During the test phase, rats were again allowed free access to both chambers without any peripheral stimulation. We compared the amount of time spent during baseline (preconditioning) and test (postconditioning) phases in each chamber (*Johansen et al., 2001*; *King et al., 2009*; *Navratilova et al., 2012*). A statistically significant reduction in the amount of time spent in a treatment chamber during the test phase when compared with baseline indicates an avoidance of that chamber, which in turn indicates aversion towards the stimulus associated with that chamber. Thus, while similar in principle to the multi-day conditioning protocols using repeated or prolonged exposure to painful stimuli to measure the acquisition of stable aversive memory (*Johansen et al., 2001*; *King et al., 2009*; *Navratilova et al., 2012*), our paradigm allows the assessment of acute aversive reaction.

First, we compared the aversive responses towards HS and NS. During conditioning, one of the chambers was paired with HS, and the other was paired with NS. We found that rats spent less time in the chamber paired with HS treatment during the test phase than at baseline (*Figure 1C*). Conversely, rats spent more time in the NS chamber during the test phase than at baseline. These results indicate that rats recognized and sought to avoid the aversive value associated with HS. Next, we measured the ability of the rats to distinguish between the aversive values of LS and NS. We conditioned rats by pairing LS with one chamber and NS in the other. After conditioning, rats spent less time in the chamber paired with LS relative to baseline, and more time in the chamber paired with NS (*Figure 1D*), indicating an ability to recognize the aversive value for LS. Finally, we conditioned rats with HS and LS, and during the test phase rats spent less time in the HS-paired chamber (and more time in the LS-paired chamber) than at baseline, suggesting that these rats displayed a greater aversive response to HS than LS (*Figure 1E*). Together, these results indicate that rats can detect not only the presence of acute pain, but also the aversive values of higher vs lower-intensity noxious stimuli. This quantitative distinction in aversive response correlates well with changes in nocifensive spinal reflex (*Figure 1B*).

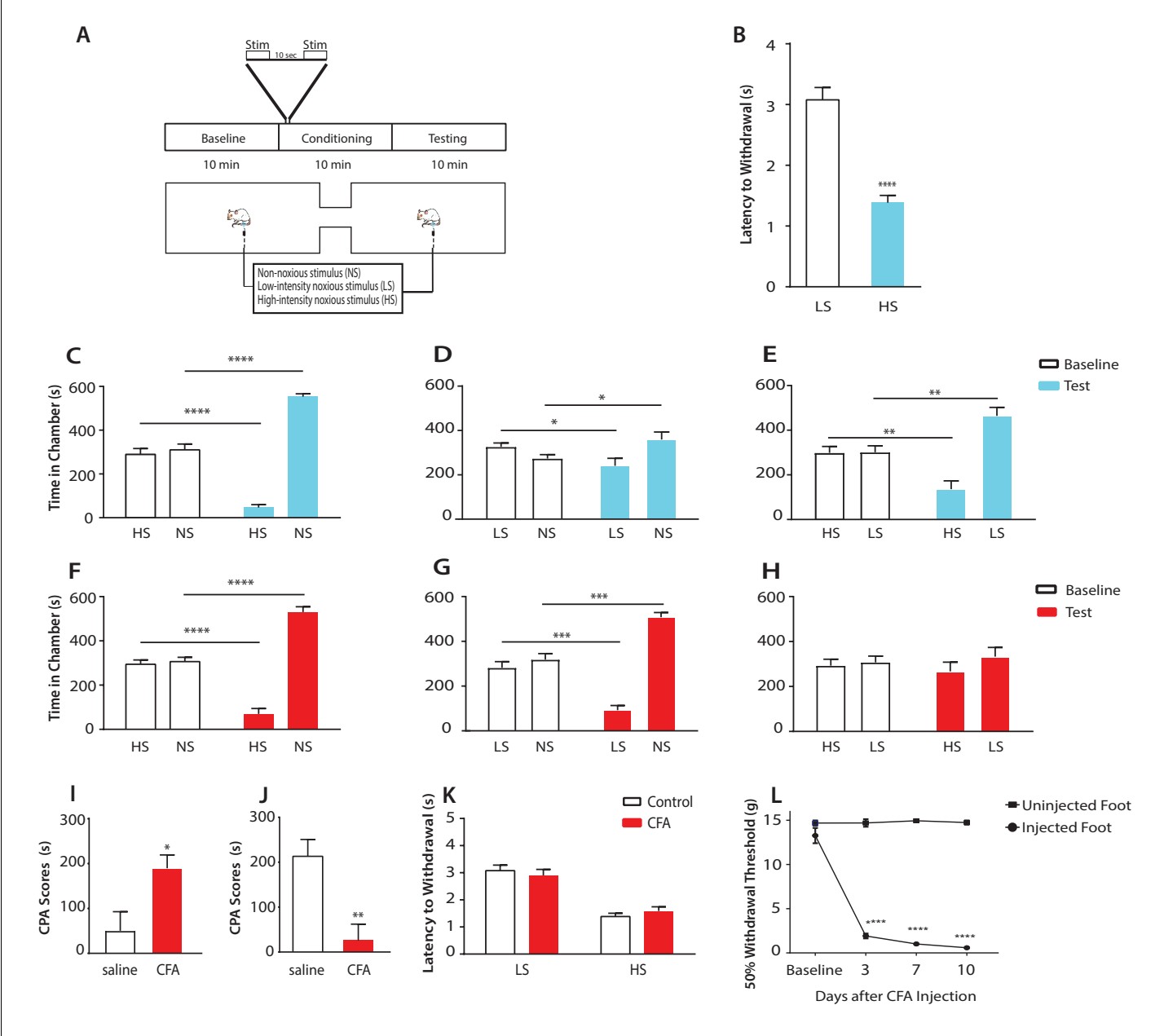

**Figure 1.** Chronic pain enhances the aversive response to acute pain at anatomically unrelated sites. (**A**) Schematic for the conditioned place aversion (CPA) test. Each episode of peripheral stimulation lasted until paw withdrawal or in cases of no withdrawal (non-noxious stimulus or NS) a total of 5 s. (**B**) Latency to paw withdrawal was shorter with higher intensity pain stimulus (HS) than lower intensity stimulus (LS). n = 12; p<0.0001, Student's t test. (**C**) During conditioning, rats received HS in one chamber and NS in the other chamber. After conditioning, rats spent less time in the chamber paired with HS during the test phase than at baseline, and more time in the chamber paired with NS. n = 14; p<0.0001, paired Student's t test. (**D**) After conditioning with LS and NS in separate chambers, rats spent less time in the chamber paired with LS during the test phase than at baseline, and more time in the chamber paired with NS. n = 14; p=0.0131. (**E**) After conditioning with HS and LS, rats spent less time in the chamber paired with HS during the test phase than at baseline, and more time in the chamber paired with LS. n = 10; p=0.0012. (**F**) 10 days after CFA treatment, rats underwent conditioning by receiving HS stimulation of the uninjected paw in one chamber and NS stimulation of the same paw in the other chamber. During the test phase, rats spent less time in the chamber paired with HS than at baseline, and more time in the chamber paired with NS. n = 9; p<0.0001; paired Student's t test. (**G**) After CFA treatment, after conditioning with LS and NS in separate chambers, rats spent less time in the chamber paired with LS stimulation of the uninjected paw during the test phase than at baseline, and more time in the chamber paired with NS. n = 9; p=0.0002. (**H**) After CFA treatment, rats could not differentiate between HS and LS treatments of the uninjected paws on the CPA test. n = 13; p=0.4923. (**I**) CFA treatment resulted in increased aversive response to LS. Rats were conditioned with LS and NS, and the CPA or aversion score was calculated by subtracting the amount of time rats spent during the test phase from baseline in the LS-paired chamber (see Materials and methods section). CPA scores for CFA-

*Figure 1 continued on next page*

*Figure 1 continued*

treated rats were significantly higher than saline control. n = 9–10; p=0.0179, unpaired Student's t test. (**J**) After CFA treatment, rats could not distinguish between HS and LS. Rats were conditioned with HS in one chamber and LS in the other chamber. The CPA score for HS was calculated by subtracting the amount of time spent in the HS-paired chamber during the test phase from baseline. CFA-treated rats demonstrated a significant decrease in the CPA score, indicating an inability to distinguish between the aversive values of HS and LS. n = 8–13; p=0.0029, unpaired Student's t test. (**K**) CFA treatment did not alter paw withdrawal latency in the uninjected paw. n = 12; p=0.8381, two-way ANOVA with repeated measures and *post-hoc* Bonferroni test. (**L**) CFA treatment caused mechanical allodynia in the injected but not uninjected paw. n = 12; p<0.0001. two-way ANOVA with repeated measures and *post-hoc* Bonferroni test.

The following figure supplement is available for figure 1:

**Figure supplement 1.** HS, LS and NS trigger high-intensity, low-intensity noxious and non-noxious stimulus respectively.

Next, we tested if this quantifiable aversive response to acute pain is altered by the presence of chronic pain at an anatomically unrelated location. We injected Complete Freund's Adjuvant (CFA) to induce persistent inflammatory pain in the opposite paw and performed CPA by conditioning with noxious stimulation of the uninjected paw. 10 days post-CFA injections, rats were conditioned with HS and NS to the healthy paw, and they demonstrated a preference for the NS chamber and avoidance of the HS chamber during the test phase compared with baseline (*Figure 1F*). These results suggest that rats in chronic pain were still able to distinguish between highly noxious from non-noxious stimulations. We then examined the response of rats in chronic pain to the low-intensity noxious stimuli. First, CFA-treated rats were conditioned with LS and NS in separate chambers. During the test phase these rats spent less time in the LS chamber than at baseline, indicating an aversive response to LS (*Figure 1G*). Interestingly, CFA-treated rats appeared to demonstrate an increased avoidance of the LS-paired chamber than rats that did not have chronic pain (compare *Figure 1G* with *Figure 1D*). We then examined the ability for these CFA-treated rats to distinguish between the aversive quality of LS vs HS. We found that CFA-treated rats did not seem able to clearly distinguish between LS and HS, as they failed to avoid the HS chamber after conditioning (*Figure 1H*), a striking difference from the avoidance of the HS chamber exhibited by rats that did not experience chronic pain (*Figure 1E*). These comparisons suggest qualitatively that while rats in chronic pain were still able to distinguish between non-noxious and noxious stimuli, they seemed to have developed a heightened aversive response to the low-intensity stimuli even at an anatomically distinct site.

To provide a quantitative analysis for the above findings, we calculated a CPA score (or aversion score) to measure the aversive value of peripheral stimulations. This CPA score was computed by subtracting the amount of time rats spent in a chamber paired with a specific noxious stimulus during the test phase from the time they spent in that chamber at baseline (*Johansen et al., 2001*; *Johansen and Fields, 2004*; *De Felice et al., 2013*). A higher CPA score (a greater difference in the time spent in the associated chamber between baseline and test phase) indicates a greater aversive value for that stimulus. We first calculated the CPA score for LS, after we conditioned rats with LS in one chamber and NS in the other. We found that this CPA score was significantly higher for CFA-treated rats than saline-treated (control) rats (*Figure 1I*). This quantitative analysis confirms that CFA-treated rats avoided the LS chamber more than control rats, and thus rats in chronic pain developed an increased aversive response to acute low-intensity noxious stimulations even at an anatomically distinct site. Next, we quantified the ability for the rats to distinguish between HS and LS. This time, we conditioned rats with HS and LS, and we computed the CPA score for HS, by subtracting the amount of time rats spent in the HS chamber during the test phase from the time they spent at baseline. In this case, a higher CPA score indicates a stronger aversive response to HS, and hence a greater ability to distinguish between HS and LS. Here, we found that CFA-treated rats demonstrated a dramatic reduction in their CPA score (*Figure 1J*), indicating that rats in chronic pain lost the ability to distinguish between the aversive values of HS and LS. Results from *Figure 1I* together indicate that after chronic pain, rats perceived both low-intensity and high-intensity noxious stimuli as highly aversive. Thus, the presence of chronic pain induced an alteration in the aversive evaluation of acute pain signals even at an anatomically distinct site. There have been reports suggesting that unilateral noxious stimulations can in some cases cause bilateral changes in spinal circuits regulating mechanical sensitivity (*Gao and Ji, 2010*, *Gao et al., 2010*). We tested whether the healthy paws in

our study displayed any behavioral signs of spinal or peripheral hypersensitivity as the result of CFA injection in the opposite paws, but we did not find any abnormality in the withdrawal reflex to noxious stimulation or the presence of mechanical allodynia (*Figure 1K*). Thus, the effect of chronic pain on acute pain responses at anatomically unrelated sites is likely specific for the aversive component, sparing the sensory component. We termed this anatomically nonspecific increase in the aversive response to acute pain 'generalized enhancement of pain aversion.'

## Chronic pain disrupts the ACC representation of acute pain signals

Having established this phenomenon of generalized enhancement of pain aversion, we investigated whether a disruption in the ACC representation of acute pain contributes to the neural substrate for such behavior. We conducted extracellular recordings in the ACC in freely moving rats before, during and after peripheral stimulation by NS, LS or HS, and analyzed the firing rates of individual neurons (*Figure 2A*). We found that a number of neurons in the ACC increased their firing rates after an acute pain stimulus (see Materials and methods; *Figure 2C*), similar to reports from human studies (*Hutchison et al., 1999*). We also found a significant number of neurons that responded to noxious stimulation and at the same time showed increased firing rate after HS than LS (*Figure 2D,E*). The identification of such neurons that responded to noxious intensity is also compatible with previous reports (*Sikes and Vogt, 1992*; *Yamamura et al., 1996*; *Hutchison et al., 1999*; *Kung et al., 2003*; *Iwata et al., 2005*; *Kuo and Yen, 2005*; *Zhang et al., 2011*). In order to make an unbiased assessment of the contribution of individual ACC neurons in pain representation, we then applied a population-decoding analysis using a support vector machine (SVM) classifier (see Materials and methods). We analyzed groups of neurons across multiple recording sessions, each session comprising of >30 trials. We used some of the trials in training and the remaining trials for testing. Both pain-responsive and non-responsive neurons were used in unbiased decoding analysis, but the firing activity from pain-responsive neurons would contribute to a higher weight (i.e., being the 'support vector'; see Materials and methods). Our decoding analysis yielded high accuracy in distinguishing between NS and HS (85% accuracy) or between LS and HS (76% accuracy) (*Figure 2F*, *Figure 2—figure supplement 1*). Therefore, our unbiased decoding analysis supported a critical role of ACC neurons in the representation of pain intensity.

Our neurophysiological recordings were performed in freely moving animals, and thus baseline movement was unlikely to affect our data interpretation. However, noxious stimulation also induced paw withdrawals, and these movements are spinal reflexes in nature (*Vardeh et al., 2016*). To ensure that these spinal reflexes did not influence our neural findings, we quantitatively analyzed these motor responses. We found that while rats did not withdraw their paws in response to NS (<5%), the percentages of withdrawal responses to LS and HS were equal (~100%). To quantify the motor function of paw withdrawals, we then calculated the withdrawal velocity after LS and HS. We did not find any statistical difference in the velocity of withdrawals (*Figure 2—figure supplement 2*). These results indicate that there was no significant difference in the gross motor responses to LS and HS, in contrast to the dramatic difference in neural responses in the ACC seen in *Figure 2*. Thus, the neural responses we observed were less likely to be related to stimulus-induced movement. Such responses in the ACC more likely represented nociceptive processing, compatible with previous reports (*Sikes and Vogt, 1992*; *Hutchison et al., 1999*; *Wang et al., 2003*; *Kuo and Yen, 2005*; *Zhang et al., 2011*).

Next, we examined how the presence of chronic pain in the opposite limb affected ACC representation of acute pain (*Figure 2G*). After CFA injection in the opposite foot, we continued to find neurons in the ACC that responded to pain stimuli (*Figure 2H*). In addition, our SVM decoding algorithm continued to distinguish HS from NS (80% accuracy) and HS from LS (67% accuracy) (*Figure 2J*) – as compared to the chance level of 50% (*Figure 2—figure supplement 3*). We then examined the 'tuning curves' of all ACC neurons that showed increased peak firing rates after HS than LS. Such tuning curves demonstrate the degree of neural responsiveness to different intensities of noxious stimulation. In the chronic pain condition, however, neurons showed a flatter tuning curve, as indicated by a significant decrease in the slope of linear fit (*Figure 2K*). This suggests a decreased ability to distinguish between LS and HS at the level of individual neurons. Furthermore, our unbiased machine learning analysis also indicated that chronic pain caused a statistically significant decrease in the ability to decode the difference between LS and HS signals (control vs. CFA: 76% vs 67% accuracy, *Figure 2L*). Thus, our neural data indicate a disruption of ACC representation

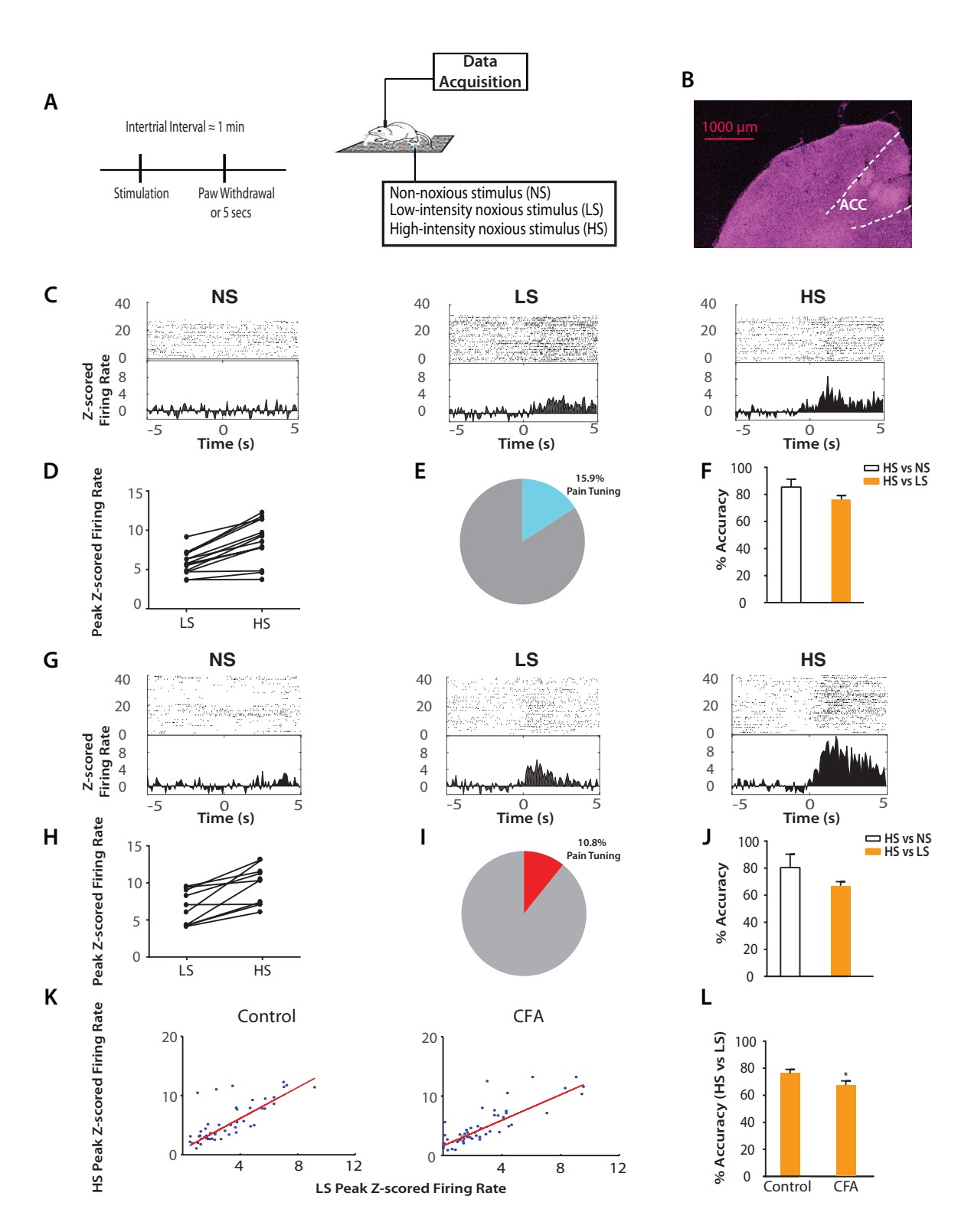

**Figure 2.** Chronic pain increases the responsiveness of ACC neurons to acute pain signals. (**A**) Timeline and schematic for electrophysiological recordings in freely moving rats. Each trial of peripheral stimulation lasted until paw withdrawal or in cases of no withdrawal (non-noxious stimulus or NS) a total of 5 s. (**B**) Histology showing the location of ACC tetrode recordings. (**C**) Raster plots and peristimulus time histograms (PSTHs) before and after NS, LS and HS stimulations. Time zero denotes the onset of stimulus. Y-axis shows z-scored firing rates. To calculate the z-scored firing rate, we

*Figure 2 continued on next page*

*Figure 2 continued*

used the following equation: (FR – mean of $FR_b$) / SD of $FR_b$, where FR indicates firing rate and $FR_b$ indicates baseline firing rate prior to NS, LS or HS (see Materials and methods section). (D, E) A subset of neurons was identified among pain responsive neurons that showed higher firing rates after HS stimulation compared with LS (see Materials and methods for definition of pain responsive neurons). n = 14 out of a total of 88 neurons; for D, p<0.0001, paired Student's t test. (F) Population-decoding analysis using a SVM classifier demonstrated decoding accuracy to distinguish between HS and NS (85%) or HS and LS (76%), n = 9. See Materials and methods. (G) Raster plots and PSTHs before and after NS, LS and HS stimulations in rats 10 days after CFA injections in the opposite paws. (H, I) A subset of neurons was found among pain responsive neurons that showed higher firing rates after HS stimulation compared with LS in the chronic pain condition. n = 10 out of a total of 93 neurons; for H, p<0.0001, paired Student's t test. (J) Population-decoding analysis demonstrated decoding accuracy to distinguish between HS and NS (81%) or HS and LS (67%) in rats after chronic pain, n = 15. (K) A robust linear regression model was used to fit the peak z-scored firing rates and to calculate slope of the fit for all ACC neurons that demonstrated higher firing rates at HS compared with LS. Slope = 1.31 ± 0.09, $R^2$ = 0.5937, n = 50 neurons. After chronic pain, neurons with higher firing rates at HS than LS showed a flatter tuning curve between LS and HS responses (see Materials and methods). Slope = 1.07 ± 0.07, $R^2$ = 0.6904, n = 53 neurons. The two slope parameters are statistically different (p<0.05, unpaired Student's t-test). (L) Decoding analysis showed that after chronic pain, there was a decrease in decoding accuracy to distinguish between HS and LS, n = 9 for pre-CFA, 15 for post-CFA; p<0.05, unpaired Student's t test.

The following figure supplements are available for figure 2:

**Figure supplement 1.** Population-decoding analysis using a shorter 3 s time window.

**Figure supplement 2.** LS and HS trigger similar paw withdrawal movements.

**Figure supplement 3.** A representative example of cumulative decoding accuracy curves using the SVM classifier.

of acute pain intensity in the chronic pain state. This impaired distinction between LS and HS by ACC neurons correlates very well with behavioral findings that showed increases in the aversive valuation of LS in the chronic pain state (*Figure 1G–J*), suggesting that the ACC likely plays a role in the impact of chronic pain on the acute pain experience.

## Chronic pain impairs the bidirectional regulation of acute pain by the ACC

To define the function of ACC in generalized enhancement of pain aversion, we temporally paired noxious stimuli with optogenetic control of ACC neurons (*Figure 3A*). First, we used channelrhodopsin (ChR2) to activate ACC pyramidal neurons, by expressing ChR2 linked to a CAMKII promotor in an AAV vector in the ACC region (*Figure 3B*; *Figure 3—figure supplement 1*). We did not observe any changes in paw withdrawal latency with optogenetic stimulation (*Figure 3C*), suggesting that brief ACC activation did not strongly modulate the acute nocifensive reflex. Next, we conditioned rats by coupling ACC activation with LS in one of the CPA chambers, and LS alone in the opposite chamber during the CPA test (*Figure 3A*). We found that ACC activation during the presentation of LS increased the aversive response to LS, as shown by an avoidance of the chamber associated with optogenetic activation during the test phase compared with baseline (*Figure 3D*). This pain-enhancing effect of ACC activation was not observed when the animal was presented with NS stimulation (*Figure 3E*), indicating that while transient ACC activation does not turn an acute non-noxious stimulus into a noxious one, it can significantly enhance the aversive quality of a noxious stimulus. Meanwhile, ACC activation during the presentation of HS increased the aversive response to HS on the CPA, but this effect was not statistically significant (*Figure 3F*). There are two possible interpretations for this apparent lack of effect of ACC activation during HS. First, our CPA test serves as a pain-scale report for rats, and with highly noxious stimulation (HS), there may be a maximal aversive response expressed by rats, similar to a maximal pain score experienced by humans. A second possible interpretation is that there may be a limit (ceiling effect) on the aversive response that could be elicited with our CPA assay, and the further aversive effect of ACC activation on HS was limited by such constraints. Nevertheless, our results clearly indicate that ACC activation during the presentation of a low intensity noxious input is capable of elevating the aversive value of that input, demonstrating that the ACC can indeed regulate the behavioral response to acute pain signals.

To ensure that activation of the ACC did not impact movement of the animals, we performed a locomotion test and found no changes in locomotor activity after ACC activation (*Figure 3—figure*

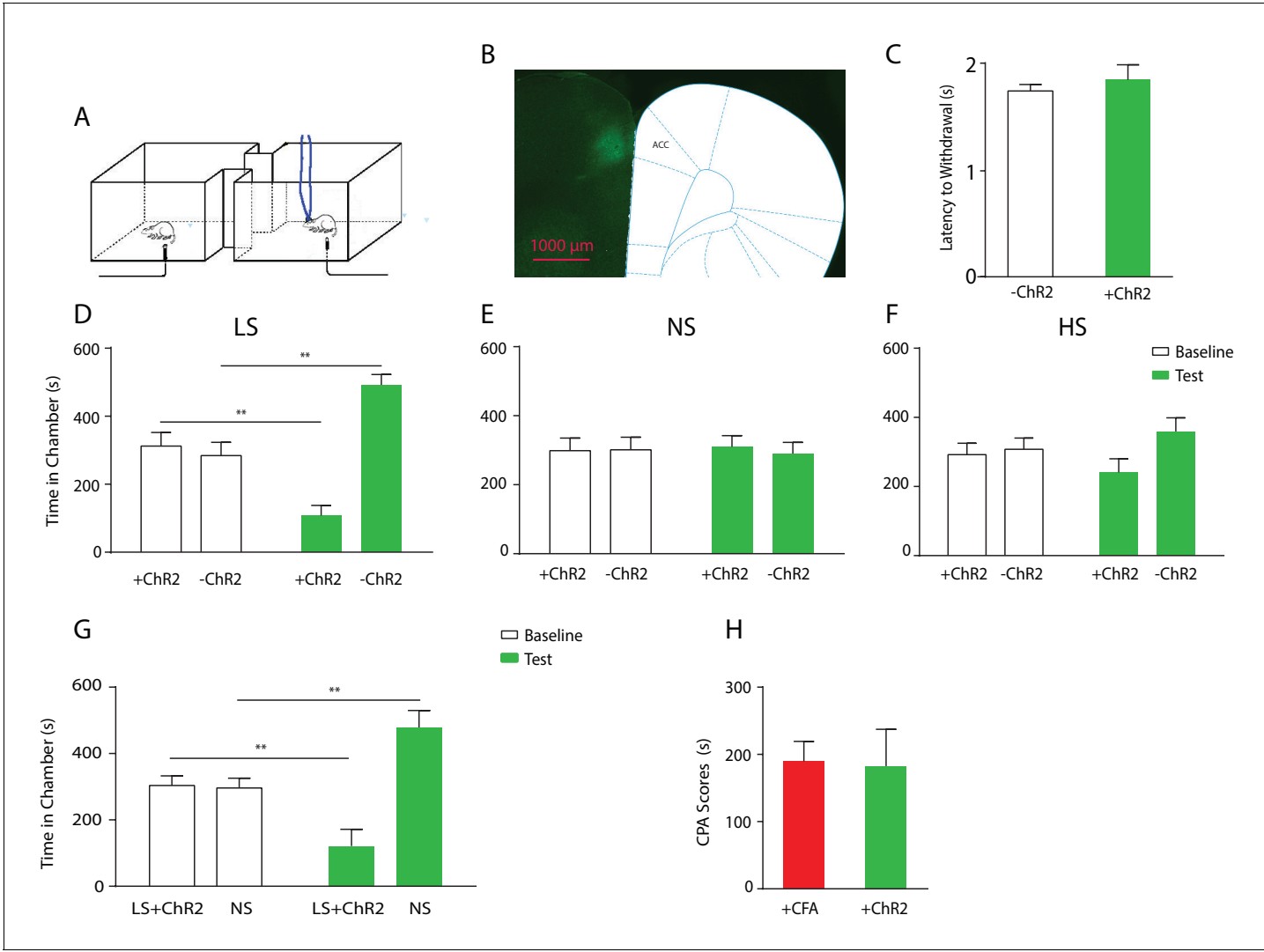

**Figure 3.** Optogenetic activation of the ACC has similar effects as chronic pain in enhancing the aversive response to acute pain. (**A**) Schematic for a CPA test during optogenetic activation of the ACC. Light activation of the ACC was temporally coupled with peripheral stimulation to the paw. (**B**) Histologic expression of ChR2 in the ACC. (**C**) Light activation of the ACC did not alter paw withdrawal latency to noxious stimuli. n = 5; p=0.4654, paired Student's t test. (**D**) ACC activation increased the aversive response to LS. One of the chambers was paired with optogenetic stimulation of the ACC and LS; the other chamber was paired with LS without ACC activation. Rats spent less time during the test phase than at baseline in the chamber paired with LS coupled with ACC activation and more time in the chamber paired with LS alone. n = 8; p=0.0026. (**E**) ACC activation did not elicit an aversive response to NS. One of the chambers was paired with ACC activation and NS; the other chamber was paired with NS alone. n = 9; p=0.7514. (**F**) ACC activation did not increase the aversive response to HS in a statistically significant manner. One of the chambers was paired with ACC activation and HS; the other chamber was paired with HS alone. n = 10; p=0.1584. (**G**) Coupling with ACC activation increased the aversive response of LS compared with NS. One of the chambers was paired with optogenetic stimulation of the ACC and LS; the other chamber was paired with NS alone. Rats spent significantly less time during the test phase than at baseline in the chamber paired with ACC activation and LS. n = 11; p=0.0075. (**H**) ACC activation caused a similar increase in the aversive response to LS as chronic pain. For comparison, rats were conditioned with LS and NS in separate chambers in both experiments. In one experiment (**3G**), LS was coupled with ACC activation, and the CPA score was calculated by subtracting the amount of time spent during the test phase from baseline in the chamber paired with simultaneous ACC activation and LS. In the second experiment, conditioning was performed in CFA-treated rats without ACC activation. A CPA score was calculated by subtracting the amount of time spent during the test phase from baseline in the chamber paired with LS in CFA-treated rats. The CPA scores from these two different experiments were similar. n = 9–11; p=0.9161, unpaired Student's t test.

The following figure supplements are available for figure 3:

**Figure supplement 1.** Expression of YFP-ChR2 in the ACC.

**Figure supplement 2.** Activation of the ACC did not alter locomotion.

*Figure 3 continued on next page*

*Figure 3 continued*

**Figure supplement 3.** Light treatment of the ACC injected with an YFP-only vector did not impact conditioned place aversion.

*supplement 2*). Furthermore, as expected, shining light in the ACC injected with a YFP-only vector without opsin expression did not result in any changes in pain aversion phenotypes (*Figure 3—figure supplement 3*). These control experiments support the specific role of ACC neurons in the aversive response to pain.

To further understand the role of the ACC in generalized enhancement of pain aversion, we compared the effect of ACC activation with CFA treatment on the aversive evaluation of noxious stimulations. First, we conditioned rats by coupling ACC activation with LS in one chamber, and NS without ACC modulation in the opposite chamber. This experiment allowed us to assess the aversive value of LS during the activation of ACC relative to a non-noxious condition. When we compared the avoidance of the LS chamber in these rats that received ACC activation (*Figure 3G*) with naïve rats (*Figure 1D*), we noticed that ACC activation likely caused an additional increase in the aversive value of LS. To quantify this increase, we measured the CPA score for LS in this experiment by subtracting the amount of time rats spent during the test phase from baseline in the chamber paired with LS and ACC activation. This CPA score represents the aversive value for the LS in the presence of ACC activation. Next, we computed the aversive value for LS in the chronic pain state by calculating the CPA score in the experiment where we conditioned CFA-treated rats with LS vs NS without optogenetic modulation of the ACC. This second CPA score indicates the aversive value for the LS in the presence of chronic pain. We found that these two CPA scores were nearly identical (*Figure 3H*). Thus, the amplitude of the additional aversive effect provided by ACC activation during the presentation of LS is similar to the effect of chronic pain. This similarity supports the role for ACC activation in the generalized enhancement of pain aversion.

To further confirm the role of the ACC in pain regulation, we tested whether inhibition of ACC neurons could decrease the aversive response to noxious stimuli. We used halorhodopsin (NpHR) to inhibit ACC neurons during the presentation of noxious stimulation (*Figure 4A*; *Figure 4—figure supplement 1*). We conditioned rats by coupling ACC inhibition with a peripheral stimulus in one of the CPA chambers, and that stimulus alone in the opposite chamber (*Figure 4A*). We found that ACC inhibition decreased the aversive response to LS and HS, as shown by increased amounts of time spent in the chamber associated with LS or HS coupled with ACC inhibition during the test phase compared with baseline (*Figure 4C,D*). Inhibition of the ACC did not impact free movements of the animals (*Figure 4—figure supplement 2*). Furthermore, as expected, ACC inhibition had no effect on the aversive response to NS, which was not noxious (*Figure 4—figure supplement 3*). Next, we wanted to know if ACC inhibition could remove the aversive response to a noxious stimulus altogether. We paired one chamber with ACC inhibition and LS, and another chamber with NS. After conditioning, we found that ACC inhibition coupled with LS stimulation removed the avoidance of the LS-paired chamber (*Figure 4E*, p>0.05). Similarly, ACC inhibition during HS removed the higher aversive value of HS compared with LS, by eliminating the avoidance of HS-paired chamber (*Figure 4F*, p>0.05). These results indicate that ACC inhibition during the presentation of a more noxious stimulus can eliminate the greater aversive reaction towards that stimulus. Therefore, the ChR2 and NpHR data together demonstrate that neurons in the ACC can bidirectionally control the aversive response to acute pain.

Finally, we tested the effect of ACC inhibition specifically on the generalized enhancement of pain aversion. We have shown that CFA-treated rats demonstrated an increased aversive response to LS in the uninjected paw (*Figure 1G*). Here, we found that ACC inhibition during LS eliminated the avoidance of LS-paired chamber in CFA-treated rats when these rats were conditioned with NS (*Figure 4G*). Thus, ACC inhibition blocked the aversion-amplifying effect of chronic pain. Furthermore, rats in chronic pain have been shown to lose the ability to differentiate between LS and HS on the CPA test (*Figure 1H*). ACC inhibition during LS stimulations, however, restored the normal aversive scale in CFA-treated rats, by reinstating the avoidance of HS chamber (*Figure 4H*). To quantitatively confirm these findings, we calculated the CPA score for CFA-treated rats which were

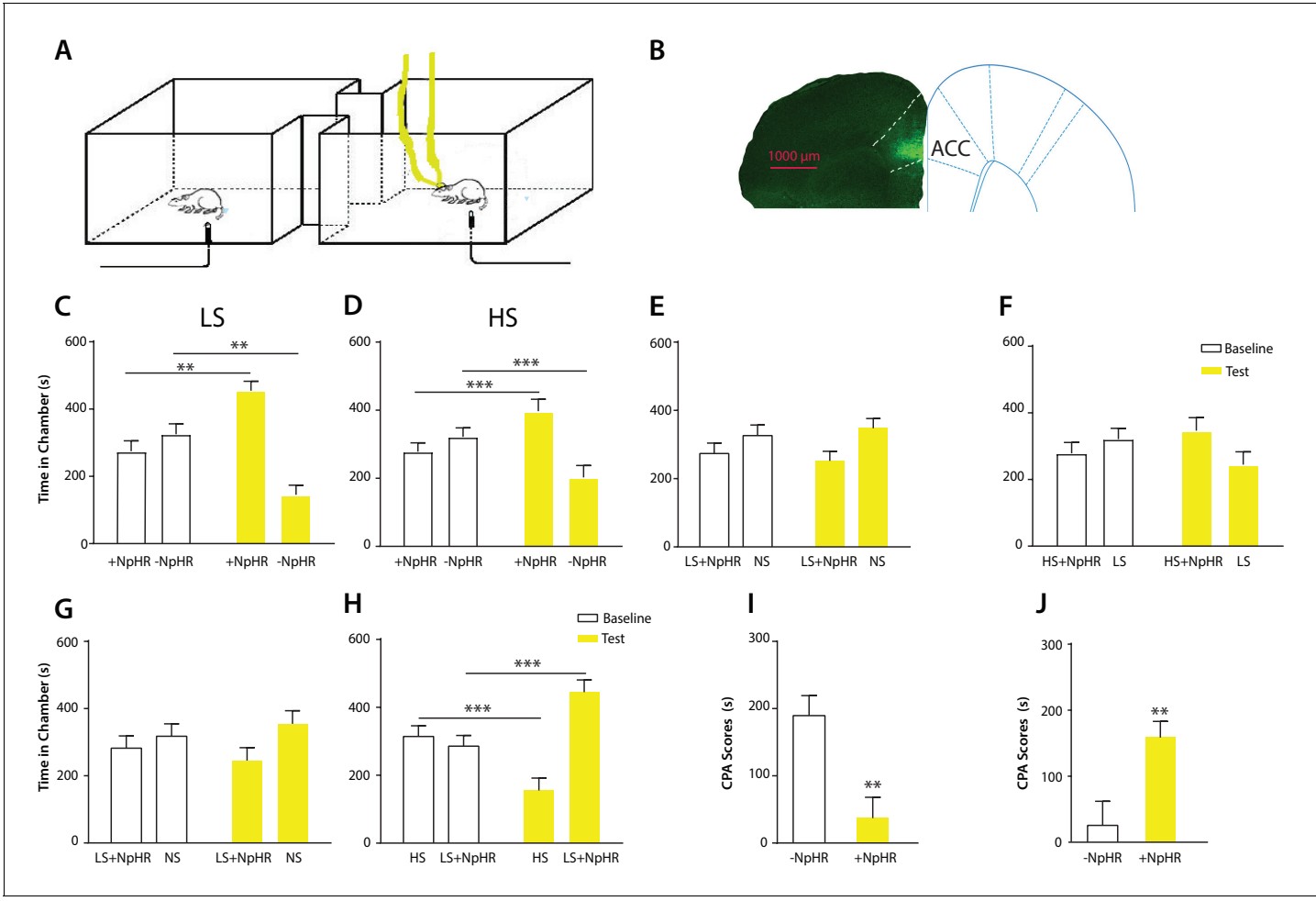

**Figure 4.** Optogenetic inhibition of the ACC diminishes the effect of chronic pain on the aversive response to acute pain. (A) Schematic for a CPA test during optogenetic inhibition of the ACC. Optogenetic inhibition of the ACC was temporally coupled with peripheral stimulation to the paw. (B) Histologic expression of NpHR in the ACC. (C) ACC inhibition decreased the aversive response to LS. One of the chambers was paired with optogenetic inhibition of the ACC and LS; the other chamber was paired with LS alone. Rats spent more time during the test phase than at baseline in the chamber paired with light inhibition of the ACC coupled with LS and less time in the LS-alone chamber. n = 10; p=0.0043, paired Student's t test. (D) ACC inhibition decreased the aversive response to HS. One of the chambers was paired with ACC inhibition and HS; the other chamber was paired with HS alone. Rats spent more time during the test phase than at baseline in the chamber paired with ACC inhibition coupled with HS and less time in the HS-alone chamber. n = 11; p=0.0006. (E) ACC inhibition abolished the aversive value of LS. One chamber was paired with ACC inhibition coupled with LS; the other chamber was paired with NS without ACC modulation. There was no statistically significant difference between test and baseline preference for either chamber. n = 9; p=0.5797. (F) ACC inhibition abolished the difference in aversive valuation between HS and LS. One chamber was paired with ACC inhibition coupled with HS; the other chamber was paired with LS. There was no statistically significant difference between test and baseline preference for either chamber. n = 12; p=0.1617. (G) ACC inhibition abolished the aversive value of LS even in CFA-treated rats. One chamber was paired with ACC inhibition coupled with LS; the other chamber was paired with NS. There was no statistically significant difference between test and baseline preference for either chamber. n = 10; p=0.2638. (H) ACC inhibition during the presentation of LS restored the normal differentiation between the aversive values of HS and LS in CFA-treated rats. One chamber was paired with ACC inhibition coupled with LS; the other chamber was paired with HS. Rats spent more time during the test phase than baseline in the chamber paired with ACC inhibition and LS, and less time in the HS-paired chamber. n = 10; p=0.0001. (I) ACC inhibition decreased the aversive value of LS in CFA-treated rats. CPA scores were compared in CFA-treated rats. In the control group (− NpHR), LS was not coupled with ACC inhibition during conditioning. In the test group (+ NpHR), LS was coupled with optogenetic inhibition of the ACC. In both groups, LS was conditioned against NS. CPA scores were calculated by subtracting the amount of time spent in the chamber paired with LS during the test phase from baseline. ACC inhibition reduced the CPA score for LS. n = 9–10; p=0.0026, unpaired Student's t test. (J) ACC inhibition restored the normal difference in the aversive response to HS vs LS even after CFA treatment. In the control group (− NpHR), LS was not coupled with ACC inhibition during conditioning. In the test group (+ NpHR), LS was coupled with optogenetic inhibition of the ACC. In both groups, LS was conditioned against HS. CPA scores were calculated by subtracting the amount of time spent during the test phase in the chamber paired with HS from baseline. ACC inhibition increased the CPA score for HS. n = 10–13; p=0.0097.

*Figure 4 continued on next page*

*Figure 4 continued*

The following figure supplements are available for figure 4:

**Figure supplement 1.** Expression of YFP-NpHR in the ACC.

**Figure supplement 2.** Inhibition of the ACC did not alter locomotion.

**Figure supplement 3.** Coupling of light inhibition of the ACC with non-noxious stimulation (NS) did not result in aversion.

conditioned with LS with or without ACC inhibition and NS. This CPA score was computed by subtracting the amount of time rats spent during the test phase from baseline in the chamber paired with LS (*Figure 4I*). We found that ACC inhibition reduced this CPA score and hence the aversive response for LS even in rats with chronic pain. We also calculated the CPA score for CFA-treated rats which were conditioned with HS and LS in the presence or absence of ACC inhibition, by subtracting the amount of time rats spent during the test phase from baseline in the chamber paired with HS (*Figure 4J*). We found that ACC inhibition during LS elevated the CPA score for HS and hence reinstated the difference in aversive values between HS and LS in rats with chronic pain. Thus, ACC inhibition restored the normal aversive response to acute pain in those rats with chronic pain. These results strongly indicate that neural activities in the ACC play a vital role in the generalized enhancement of aversion in chronic pain conditions.

## Discussion

The key finding in our study is that chronic pain causes a generalized enhancement in pain aversion. Previous studies have revealed the aversive state secondary to nociceptive inputs from the site of chronic pain (*Johansen et al., 2001*; *Johansen and Fields, 2004*; *King et al., 2009*; *De Felice et al., 2013*). The novel aspect of our results lies in the demonstration that chronic pain at one site in the body can also increase the aversive response to acute pain in a separate location. Interestingly, acute pain responses elicited by lower or intermediate intensity stimulus (such as LS) were more affected by chronic pain than maximal pain stimulus (HS), leading to a distortion of the pain-intensity scale. Epidemiological studies have shown that the presence of chronic pain coincides with increased pain severity and a similar distortion of pain intensity scale in a diffuse anatomic pattern (*Scudds et al., 1987*; *Petzke et al., 2003*; *Kehlet et al., 2006*; *Kudel et al., 2007*; *Scott et al., 2010*). Our study confirms these clinical findings by providing a causal link between chronic pain and a generalized enhancement of pain aversion. This generalized enhancement in aversion may be an important mechanism for chronic pain to influence normal sensory and affective processes.

Our study also provides a mechanistic basis for this generalized enhancement of pain aversion. Individual ACC neurons can respond to noxious stimuli by increasing firing rates (*Sikes and Vogt, 1992*; *Yamamura et al., 1996*; *Hutchison et al., 1999*; *Kung et al., 2003*; *Iwata et al., 2005*; *Kuo and Yen, 2005*; *Zhang et al., 2011*). Our study shows that this neural representation of acute pain intensity is profoundly altered by chronic pain, which causes ACC neurons to display a disproportional response to low-intensity pain stimuli. Previous studies have found that chronic pain can induce maladaptive synaptic plasticity in the ACC (*Li et al., 2010*; *Koga et al., 2015*). It is possible that such plasticity increases the response of ACC neurons to acute pain signals, and responses to low-intensity stimuli are disproportionally affected because the neuronal response is possibly saturated at high-intensity stimuli in the absence of chronic pain. At the network level, ACC neurons are known to project to or receive inputs from a number of regions important for pain processing, including the prefrontal cortex, medial thalamus, insular cortex, amygdala, nucleus accumbens, hippocampus, etc. Thus, acute pain likely triggers a concerted neural response in an interconnected network as suggested by fMRI studies (*Schweinhardt and Bushnell, 2012*; *Wager et al., 2013*). Chronic pain, however, has the capacity to disturb this network response.

Another important finding in our study is the ability of the ACC to exert bidirectional control of pain aversion. Prior lesion and pharmacological studies have shown that the rostral region of the ACC is required specifically for the acquisition of stable aversive learning induced by chronic pain (*Johansen et al., 2001*; *Johansen and Fields, 2004*; *Qu et al., 2011*). Our study indicates that this

brain region is also necessary and sufficient for the temporal regulation of the aversive response to acute pain stimuli. Interestingly, in our study, brief activation of the ACC did not cause an aversive reaction to a non-noxious stimulus, whereas previous studies demonstrate that repeated activation of ACC neurons itself can be an aversive teaching signal (*Johansen and Fields, 2004*). This difference suggests that repeated and possibly prolonged activation of ACC is required for the acquisition of stable aversive memory, whereas transient activation provides context-specific aversive valuation and response. At the molecular and cellular level, persistent peripheral nociceptive inputs have been shown to trigger opioid signaling and synaptic plasticity in the ACC to regulate sensory and aversive components of chronic pain (*Li et al., 2010*; *Navratilova et al., 2015*). Our results here suggest that these mechanisms have the potential to impact ACC regulation of pain aversion in an input- and output-nonspecific fashion to exert a more generalized form of control for acute pain behavior.

In addition to pain regulation, the anterior cingulate cortex is also involved in a number of sensory, affective and cognitive processes (*de Araujo et al., 2003*; *Rolls et al., 2003*; *Grabenhorst et al., 2008*; *Rolls et al., 2008*). It plays an important role in reward-based learning, such as providing necessary evaluation for rewarding or aversive cues as well as for the interpretation of errors in predicting such cues (*Bush et al., 2002*). It also plays roles in attention and in conflict monitoring (*Braver et al., 2001*). While our experiment demonstrated that ACC changes are likely specific to pain instead of motor responses to pain, we cannot absolutely rule out all the potential behavioral covariance associated with noxious stimulation. However, given the complexity and diversity of its functions and anatomic connections, it is perhaps not surprising that plasticity within the ACC as the result of chronic pain can alter the regulation of general aversive responses. It should be noted, in addition, that given its rich functional connectivity, the ACC is likely to be an important node in a complex network of brain structures that regulate this generalized enhancement in pain aversion.

In summary, we have demonstrated that chronic pain can disrupt cortical function to increase the aversive response to acute noxious signals in an anatomically nonspecific manner. This mechanism of generalized enhancement of pain aversion may underpin the pathophysiology of diffuse pain syndromes such as fibromyalgia and chronic postoperative pain. It also raises the possibility that other conditions such as depression and anxiety may exert similar impact on acute pain or other sensory and affective processes in general.

## Materials and methods

### Animals

All procedures in this study were approved by the New York University School of Medicine (NYU-SOM) Institutional Animal Care and Use Committee (IACUC) as consistent with the National Institute of Health (NIH) *Guide for the Care and Use of Laboratory Animals* to ensure minimal animal use and discomfort. Male Sprague-Dawley rats were purchased from Taconic Farms, Albany, NY and kept at Mispro Biotech Services Facility in the Alexandria Center for Life Science, with controlled humidity, temperature, and 12 hr (6:30 AM to 6:30 PM) light-dark cycle. Food and water were available *ad libitum*. Animals arrived to the animal facility at 250 to 300 grams and were given on average 10 days to adjust to the new environment prior to the onset of experiments.

### Complete Freund's Adjuvant (CFA) administration

To produce chronic inflammatory pain, 0.1 ml of CFA (mycobacterium tuberculosis, Sigma-Aldrich) was suspended in an oil-saline (1:1) emulsion and injected subcutaneously into the plantar aspect of the hindpaw opposite to the paw that was stimulated by laser. Control rats received equal volume of saline injection.

### Virus construction and packaging

Recombinant AAV vectors were serotyped with AAV1 coat proteins and packaged by the viral vector core at the University of Pennsylvania. Viral titers were $5 \times 10^{12}$ particles/mL for AAV1.CAMKII. ChR2-eYFP.WPRE.hGH and AAV1.CAMKII.NpHR-eYFP.WPRE.hGH.

## Stereotaxic optic fiber implantation and intracranial viral injections

As described previously (*Goffer et al., 2013*; *Lee et al., 2015*), rats were anesthetized with isoflurane (1.5% to 2%). In all experiments, virus was delivered to the anterior cingulate cortex (ACC) only. Rats were bilaterally injected with 0.5 μL of viral vectors at a rate of 0.1 μL/10 s with a 26-gauge 1 μL Hamilton syringe at anteroposterior (AP) +2.6 mm, mediolateral (ML) ±1.6 mm, and dorsoventral (DV) −2.25 mm, with tips angle 28° toward the midline. The microinjection needles were left in place for 10 min, raised 1 mm and left for another minute to allow for diffusion of virus particles away from injection site while minimizing spread of viral particles along the injection tract. Rats were then implanted with 200 μm optic fibers held in 1.25 mm ferrules (Thorlabs) in the ACC: AP +2.6 mm, ML ±1.6 mm, DV −1.25 mm. Fibers with ferrules were held in place by dental acrylic.

## Electrode implantation and surgery

Tetrodes were constructed from four twisted 12.7 μm polyimide-coated microwires (Sandvik) and mounted in an eight tetrode VersaDrive (Neuralynx). Electrode tips were plated with gold to reduce electrode impedances to 100–500 kΩ at 1 kHz. Rats were anesthetized with isoflurane (1.5–2%). The skull was exposed and a 2.5-mm-diameter hole was drilled above the target region. A durotomy was performed before tetrodes were slowly lowered unilaterally into the ACC with the stereotaxic apparatus. Coordinates for ACC implants were: AP +2.7 mm, ML 0.8 mm, and DV 1.4 mm, with tetrode tips angled 10° toward the midline. The drive was secured to the skull screws with dental cement.

Following animal sacrifice, brain sections were collected at a thickness of 20 μm using Microm HM525 Cryostat machine, and sections were analyzed for viral expression and optic fiber localization with histological staining. Animals with improper fiber or electrode placements, low viral expression, or viral expression outside the ACC were excluded from the study.

## In vivo electrophysiological recordings

Before stimulation, animals were given a 30 min period to habituate to a recording chamber over a mesh table. Noxious stimulation via a 1000 mW blue diode-pumped solid-state laser (Shanghai Dreams Laser Technology Co., LTD.) was applied 1 mm from the plantar surface of the hind paw contralateral to the brain recording site in freely moving rats (*Chen et al., 2017*). The laser output intensity could be NS (laser intensity of 50 mW), LS (150 mW) or HS (250 mW) (see below) in a single session with 200 μm core diameter fiber (M83L01, Thorlabs). The laser output power was calibrated by compact power and energy meter console (PM100D, Thorlabs) at the beginning of every recording session. In a single trial, the laser was turned on by a transistor-to-transistor (TTL) pulse generator (Doric) until paw withdrawal was observed (or for a total of 5 s if no withdrawal occurred). All recording sessions consisted of approximately 35 trials with variable inter-trial intervals (approximately 1 min) using one category of stimulation (NS, LS, or HS). A video camera (HC-V550, Panasonic) was used to record the experiment. Long inter-trial intervals between trials and the break between sessions were used to avoid sensitization. We did not identify any sensitization behavior nor physical damage to the paw during our experiment. The withdrawal latency was defined by the time between onset of laser and paw withdrawal.

In a subset of experiments intended for decoding analysis, we performed recordings using two different lasers to provide two different output intensities (NS&HS, or LS&HS). The stimulations were randomly applied to rat's hind paw for a total of approximately 60 trials (of equal number of trials for each stimulation intensity).

Optrode recordings were made when animals were anesthetized with 1% isoflurane and secured to a stereotaxic apparatus, as described previously (*Lee et al., 2015*). A 32-channel optrode (VersaDrive8 optical, Neuralynx) containing an optical fiber positioned 0.5 mm above the tips of 8 surrounding tetrodes was implanted after virus injection, using the coordinates mentioned above. The optical fiber was connected to a laser, which was connected to a TTL pulse-generating box (OTPG4, Doric Instruments). An extra output on the TTL box and the headstage were then connected to the data acquisition system in order to simultaneously record laser pulses and brain activity.

## Data collection and preprocessing

Tetrodes were lowered in steps of 120 μm before each day of recording. The neuronal activity and the onset of noxious laser stimulation were simultaneously recorded with an acquisition equipment

(Open Ephys) via an RHD2132 amplifier board (Intan Technologies). Signals were monitored and recorded from 32 low-noise amplifier channels at 30 kHz, band-passed filtered (0.3 to 7.5 kHz). To get spike activity, the raw data were high-pass filtered at 300 Hz with subsequent thresholding and offline sorting by commercial software (Offline Sorter, Plexon). The threshold was lower than the 3-Sigma peak heights line and optimized manually based on the signal to noise ratio. The features of three valley electrodes were used for spike sorting. Only clear spike clusters with good tetrode spike waveforms and ISI (inter spike interval) Poisson distribution were selected for analysis. Single units with peak firing rates lower than 1 Hz were excluded. Trials were aligned to the initiation of laser-on to compute the PSTH for each single unit using MATLAB (Mathworks).

## Immunohistochemistry

Rats were deeply anesthetized with Isoflurane and transcardially perfused with ice-cold PBS followed by 4% paraformaldehyde (PFA) in PBS. Brains were fixed in PFA overnight and then transferred to 30% sucrose in PBS to equilibrate for three days as described (*Lee et al., 2015*). 20 μm coronal sections were made with a cryostat and washed with PBS for 10 min. Sections were washed in PBS and coverslipped with Vectashield mounting medium. Images containing tetrodes were stained with cresyl violet. These images were acquired using a Nikon eclipse 80i microscope with a DS-U2 camera head. Sections were also made after viral transfer for opsin verification, and these sections were stained with anti-rabbit GFP (1:500, Abcam, Cambridge, MA, #AB290), anti-mouse VGLUT 1/2 (1:200, Millipore, Temecula, CA, #MAB5502/5504), and DAPI (1:200, Vector Laboratories, Burlingame, CA) antibodies. Secondary antibodies were anti-rabbit IgG conjugated to AlexaFluor 488, and anti-mouse IgG conjugated to AlexaFluor 647 (1:200, Life Technologies, Carlsbad, CA). Images were acquired with a Zeiss LSM 700 Confocal Microscope (Carl Zeiss, Thornwood, NY).

## Animal behavioral tests

For optogenetic experiments, optic fibers were connected to a 473 nm (for ChR2) or 589 nm (for NpHR) laser diode (Shanghai Dream Lasers) through a mating sleeve as described previously (*Lee et al., 2015*). Laser intensity was measured with a power meter (Thorlabs) prior to experiments. Laser was delivered using a TTL pulse-generator (Doric).

### Conditioned place aversion (CPA)

CPA experiments were conducted similar to what has been described previously (*Johansen et al., 2001*; *Johansen and Fields, 2004*; *King et al., 2009*; *De Felice et al., 2013*; *Lee et al., 2015*). The movements of rats in each chamber were automatically recorded by a camera and analyzed with the Any-maze software. The CPA protocol included preconditioning (baseline), conditioning, and testing phases (10 min during each phase). Animals spending more than 500 s or less than 100 s of the total time in either main chamber in the preconditioning phase were eliminated from further analysis (approximately 20% of total animals). Immediately following the pre-conditioning phase, the rats underwent conditioning for 10 min. Each of the two chambers was paired with a unique blue laser stimulus, which could be NS, LS or HS. For NS, the laser output power was approximately 50 mW, for LS, it was 150 mW, and for HS, it was 250 mW. Laser intensity was measured with a power meter (Thorlabs) prior to each experiment, and the same intensity was used consistently to generate LS and HS. A stimulus was terminated after paw withdrawal, and this occurred for LS and HS. In the case of NS, withdrawals occurred on less than 5% of stimulations, and in the case of non-withdrawals the stimulus was applied for a total of 5 s. The stimulus was repeated every 10 s. During a subset of the experiments, optogenetic activation was concurrent with laser stimulation in one of the treatment chambers. Laser stimulation, optogenetic stimulation and chamber pairings were counterbalanced. During the test phase, the animals did not receive any treatment and had free access to both compartments for a total of 10 min. Animal movements in each of the chambers were recorded, and the time spent in either of the treatment chambers was analyzed by the AnyMaze software. Decreased time spent in a chamber during the test phase as compared with the baseline indicates avoidance (aversion) for that chamber.

## Paw withdrawal latency test

A laser stimulation of 50 mW (NS), 150 mW (LS), or 250 mW (HS) was applied to the hind paw of rats. Less than 5% of the time 50 mW stimulation elicited a withdrawal response within 5 s, whereas 150 mW and 250 mW elicited withdrawal 100% of the time. A video was used to tape the procedure and analyze the withdrawal latency.

## Mechanical allodynia test

A Dixon up-down method with von Frey filaments was used to measure mechanical allodynia (*Chaplan et al., 1994*; *Bourquin et al., 2006*; *Wang et al., 2011*). Rats were individually placed into plexiglass chambers over a mesh table and acclimated for 20 min before testing. Beginning with 2.55 g, von Frey filaments in a set with logarithmically incremental stiffness (0.45, 0.75, 1.20, 2.55, 4.40, 6.10, 10.50, 15.10 g) were applied to the paws of rats. 50% withdrawal threshold was calculated as described previously (*Wang et al., 2011*).

## Measurement of the velocity of paw withdrawals

We used a high speed camera (Sony Handycam FDR-AX53) to record frame by frame the movement of the paws after noxious stimulation with LS or HS. We measured the velocity of paw withdrawal by dividing the height of paw withdrawal by the time it took for the animal to reach this height with its affected paw. An average of 10 measurements were calculated for each rat.

## Locomotion test

We recorded locomotion over 10 min for rats that received optogenetic activation or inhibition of the ACC. Either blue or yellow light was turned on for 3 s every 10 s during the locomotion test. In control rats, no light activation was provided. Total distance travelled was computed based on Any-Maze recordings.

## Statistical analysis

The results of behavioral experiments were given as mean ± S.E.M. For mechanical allodynia, a two-way ANOVA with repeated measures and *post hoc* multiple pair-wise comparison Bonferroni tests were used to compare the time spent in chamber or 50% withdrawal threshold under various testing conditions. During the CPA test, a paired Student's t test was used to compare the time spent in each treatment chamber before and after conditioning (i.e. baseline vs test phase for each chamber) (*King et al., 2009*). Decreased time spent in a chamber during the test phase as compared with the baseline indicates avoidance (aversion) for that chamber. A CPA score was computed by subtracting the time spent in the more noxious chamber during the test phase from the time spent in that chamber at baseline (*Johansen et al., 2001*; *Johansen and Fields, 2004*; *De Felice et al., 2013*). Thus, for rats that were conditioned with LS and NS, CPA for LS was computed by subtracting the time spent in the LS chamber during the test phase from the time spent in that chamber at baseline. Meanwhile, for rats that were conditioned with HS and LS, CPA for HS was computed by subtracting the time spent in the HS chamber during the test phase from the time spent in that chamber at baseline. A two-tailed unpaired Student's t test was used to compare differences in CPA scores under various testing conditions.

For neuronal spike analysis, to define a neuron that altered its firing rate in response to a peripheral stimulus, we calculated peri-stimulus time histograms (PSTH), using a 5 s range before and after laser stimulus and a bin size of 200 ms. The baseline mean and standard deviation was calculated from the five second interval prior to stimulus. To calculate z-scored firing rate, we used the following equation: $Z = (FR - \text{mean of } FR_b) / \text{standard deviation of } FR_b$, where FR indicates firing rate and $FR_b$ indicates baseline firing rate prior to NS, LS or HS. To define a pain responsive neuron, we used the following criteria: (1) The absolute value of the z score firing rate of least two time bins after stimulation must be $\geq 2.33$; and (2) If the first criterion is passed, the lower bound z-score as defined by $(Z - Z_{SEM})$ at least two bins after stimulation must be greater than 1.645. $Z_{SEM}$ is defined by the following equations: $Z_{SEM}(\text{bin}) = FR_{SEM}(\text{bin})/\text{standard deviation of } FR_b$ (baseline/bin size), and $FR_{SEM}(\text{bin}) = (\text{standard error of FR over all laser trials})/\text{bin size}$. For ACC neurons that demonstrated increased firing rates after HS than LS, we also used a robust linear regression model to fit the peak z-scored firing rates in response to HS and LS and to calculate the slope of fit. This provided a

'tuning curve' to differentiate between HS and LS. For comparing the slopes of two regression lines, we used a Student's t-test (*Andrade and Estévez-Pérez, 2014*).

For all tests, a $p$ value<0.05 was considered statistically significant. All data were analyzed using the GraphPad Prism Version 7 software (GraphPad) and MATLAB (MathWorks).

## Population-decoding analysis using machine learning

After spike sorting, we obtained population spike trains from simultaneously recorded ACC neurons. For each single neuronal recording, we binned spikes into 100 ms to obtain spike count data in time. To simulate the online decoding, we used a 100 ms moving window to accumulate spike count statistics from laser onset (time 0) until 5 s (i.e., 50 bins). We assessed the decoding accuracy at each time bin based on the cumulative spike count statistics. Therefore, for a total of $C$ neurons, the input dimensionality ranged from $C$ (the first bin) to $50C$ (all bins). In these experiments where we randomly mixed different laser intensities (NS and HS or LS and HS), we assumed that we have $n_1$ trials under laser intensity 1, and $n_2$ trials under laser intensity 2. We split the total ($n_1 + n_2$) trials into two groups, 80% used for training, and 20% used for testing. The goal of population-decoding analysis was to classify the trial labels of different stimulation intensities (e.g., LS vs. HS) based on population spike data. We used a support vector machine (SVM) classifier (*Bishop, 2007*). The SVM is a discriminative supervised learning model that constructs the classification boundary by a separating hyperplane with maximum margin. Specifically, the SVM can map the input $\mathbf{x}$ into high-dimensional feature spaces which allows nonlinear classification.

$$y = \sum_{i=1}^{N} \alpha_i K(\mathbf{x}, \mathbf{x}_i) + b$$

where $y_i \in \{-1, +1\}$ denote the class label for the training sample $\mathbf{x}_i$ (some of which associated with nonzero $\alpha_i$ are called support vectors), $b$ denotes the bias, and $K(\bullet, \bullet)$ denotes the kernel function. We used a polynomial kernel and trained the nonlinear SVM with a sequential minimal optimization algorithm (MATLAB Machine Learning Toolbox: 'fitcsvm' function). Finally, the decoding accuracy was assessed by 5-fold cross-validation from 100 Monte Carlo simulations. We report the mean $\pm$ S. E.M. decoding accuracy.

As a control, we also computed the chance-level decoding accuracy. We randomly permuted class labels between two classes and repeated the decoding analysis. This shuffling procedure was repeated 500 times, and we reported the chance level by the averaged classification accuracy based on shuffled data with permuted labels. In theory, when the sample sizes from both classes are perfectly balanced, the chance level should be close to 50%. Based on our experimental data, we obtained a chancel-level ~51–53%.

In all population-decoding analyses (pre- and post-CFA), we only used the recording sessions with five or more simultaneously recorded ACC units, independent of the cell firing properties. For NS vs HS, we have 6 and 4 sessions in pre-CFA and post-CFA conditions, respectively. For NS vs LS, we have 5 and 4 sessions in pre-CFA and post-CFA conditions, respectively. For LS vs HS, we have 9 and 15 sessions in pre-CFA and post-CFA conditions, respectively.

## Acknowledgements

This work was supported by the National Institute of General Medical Sciences (GM102691, GM115384, JW), National Institute of Neurological Disorders and Stroke (NS100065, ZC and JW), (Bethesda, MD, USA), the National Science Foundation (IIS-1307645, Arlington, VA, ZC) and the Anesthesia Research Fund of New York University Department of Anesthesiology (New York, NY, JW).

## Additional information

### Funding

| Funder | Grant reference number | Author |
| --- | --- | --- |
| National Institute of Neurological Disorders and Stroke | NS100065 | Jing Wang Zhe Chen |

| National Science Foundation | IIS-1307645 | Zhe Chen |
| National Institute of General Medical Sciences | GM102691 | Jing Wang |
| National Institute of General Medical Sciences | GM115384 | Jing Wang |

The funders had no role in study design, data collection and interpretation, or the decision to submit the work for publication.

## Author contributions

QZ, Conceptualization, Resources, Data curation, Software, Formal analysis, Validation, Investigation, Visualization, Methodology, Writing—original draft, Writing—review and editing; TM, Conceptualization, Resources, Data curation, Software, Formal analysis, Investigation, Methodology; APT, RY, AGa, JD, Data curation, Formal analysis, Investigation, Methodology; EM, HZ, AGo, LU, Data curation, Formal analysis, Investigation; GY, Formal analysis, Supervision, Methodology; ZC, Conceptualization, Data curation, Software, Formal analysis, Supervision, Validation, Investigation, Methodology, Writing—original draft, Project administration, Writing—review and editing; JW, Conceptualization, Resources, Data curation, Software, Formal analysis, Supervision, Funding acquisition, Validation, Investigation, Visualization, Methodology, Writing—original draft, Project administration, Writing—review and editing

## Author ORCIDs

Qiaosheng Zhang, http://orcid.org/0000-0003-0485-3126
Jing Wang, http://orcid.org/0000-0003-1580-1356

## Ethics

Animal experimentation: All procedures in this study were approved by the New York University School of Medicine Institutional Animal Care and Use Committee (IACUC) as consistent and in strict accordance with the National Institute of Health (NIH) Guide for the Care and Use of Laboratory Animals to ensure minimal animal use and discomfort. The protocol (170315-01) was approved by the ethics committee at New York University School of Medicine. All surgeries were performed under isoflurane anesthesia, and every effort was made to minimize suffering.

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
