## [Decision Letter]

[Editors’ note: a previous version of this study was rejected after peer review, but the authors submitted for reconsideration. The first decision letter after peer review is shown below.]

Thank you for choosing to send your work, "Chronic pain induces generalized enhancement of pain aversion", for consideration at *eLife*. Your initial submission has been reviewed by a Senior Editor and three reviewers, one of whom is a member of our Board of Reviewing Editors. Although the work is of interest, we regret to inform you that the findings at this stage are too preliminary for further consideration at *eLife*.

While the reviewers expressed a high degree of interest in your study, several issues raised prevent us from moving forward with the manuscript in its present form. To summarize, main concerns include a lack of detailed descriptions of how each experiment was performed, including whether the same or different cohorts of rats were used for multiple measurements, and issues with how each experiment was analyzed with statistical methods; each of the three reviewers struggled with the statistical analysis, including how CPA values in Figure 1 were calculated. Reviewers also agreed that there is a missing control experiment (effects of optogenetic inhibition of the ACC on CPA by itself (or paired with NS)). Also, the degree of viral infection of ACC neurons was not measured and the cell types that are infected and express the optogenetic probes was also lacking. As for the concern noted in individual reviews regarding whether brain regions other than the ACC are involved, the consensus following discussion amongst the reviewers was that the finding of ACC involvement is of sufficient interest even in the absence of evidence implicating other brain regions. These consensus views are noted for your consideration in preparation for submission of a new manuscript to *eLife* or elsewhere.

Reviewer #1:

This study reports that chronic pain caused by CFA injected into the paw causes a generalized enhancement in pain aversion, as measured by laser stimulation of the uninjured paw of the opposite limb. The authors make clever use of CPA to examine aversive values presented by acute pain signals, delivered by the laser. They found that a submaximal acute pain behavior elicited by a low-intensity noxious laser pulse (LS) is more affected by chronic pain than is maximal pain behavior elicited by high-intensity stimulus (HS). Presumably the HS is already maximally aversive. Physiologically, the neural representation of acute pain in the ACC elicited by the LS to the uninjured paw is altered in the chronic pain state. Although I AM not an expert in the decoding analysis used to conclude that chronic pain disrupts the ACC representation of LS, I find this study to have several interesting findings.

Concerns that would need to be addressed:

In the CPA measurement, the extent to which rats avoid the chamber in which LS is delivered in the CFA group is said to be increased compared to LS treatment of the uninjured rats (Figure 1). This observation forms the basis of the entire study, and yet I am not clear on the statistical test used to draw this conclusion. In general, the way in which CPA measurements were calculated and how statistical calculations and comparisons were made are confusing. This needs clarification.

Whether alterations of central representation of LS in the chronic pain state is unique to the ACC or whether the ACC is one component of an interconnected network of brain regions similarly altered is not clear. That is, does optogenetic perturbation of any component of an interconnected network (medial thalamus, etc) lead to similar findings? Is the ACC the locus of altered neural representation of acute pain to an uninjured limb or one component of many?

Reviewer #2:

The manuscript by Zhang et al. uses a clever experimental design to uncover enhanced aversion to noxious input in the context of ongoing pain, and then uses optogenetic approaches to provide evidence for a role of the ACC for these behavioral changes. The paper has many strengths including novelty of both the approaches and the findings.

Although this study is not the first to show bilateral changes in mechanical sensitivity associated with persistent unilateral inflammation, it is the first, to my knowledge, to measure the aversive aspect of this change in a CPA test.

The authors provide intriguing correlative data supporting a role for the ACC in the responses to noxious stimulation through tetrode recordings. I do not feel fit to assess the experiments presented in Figure 2 because they are beyond my area of expertise.

The most interesting finding in this manuscript is that optogenetic activation of ACC neurons increases the aversiveness to noxious stimulation, whereas inhibition of ACC neurons decreases the aversiveness to noxious stimulation.

Major concerns.

1) I feel that the descriptions of how the data were analyzed are insufficient for me to determine whether the analyses were done correctly. For instance, many graphs show the time spent on each side of the chamber before and after conditioning treatment. In each case, the data presented in the first two columns are the mirror image of the data presented in the second two columns (since the rats are either in one chamber or the other, so this is not an independent variable.) To analyze this type of data, the authors do a 2-way ANOVA, which I find confusing. In other experiments, they calculate a CPA score, which is defined as the difference between the time spent in either treatment chamber. Again, I'm not exactly sure which numbers were subtracted to give the resulting data. I think these experiments would benefit from a clearer description as well as careful analysis by a statistical expert (not me). The authors should also clarify whether these experiments were done on the same cohort of rats, or whether naive rats were used for each experiment.

2) Conceptually, I find it hard to wrap my head around the difference between the low intensity noxious stimulus and the high intensity noxious stimulus. In each case, the rat is exposed to a laser until (and only until) the stimulus triggers a nociceptive reflex. Granted, the authors show that in CPA tests the mice can distinguish LS from HS. And they also show that a declassifying code can distinguish responses from LS and HS. Nevertheless, I feel that, in the spectrum of possible pain, the two types of stimuli would be rather similar. For this reason, I find it peculiar that the optogenetic stimulation of the ACC affects behavioral responses to LS but not HS. The reason given "HS presumably already elicited maximal ACC response, resulting in a less pronounced effect of optogenetic activation" is both unsatisfying and overinterpreted (note that there is actually no effect of optogenetic stimulation in the HS condition).

3. It seems to me that a key control is the effect of optogenetic inhibiton of the ACC on CPA by itself (or paired with NS). This control is missing.

Reviewer #3:

The report by Zhang et al. addresses an important issue in pain biology; how do discrete injuries to one portion of the body develop into wide-spread pain syndromes that cause patients to have hypersensitivity/allodynia in areas that were unaffected by the original insult? The answer to this question will have a direct and profound impact on a number of chronic pain conditions for which there are currently no treatments.

To address this issue, Zhang et al. used state of the art techniques to probe the role of the anterior cingulate cortex (ACC) in modulating the sensitivity of the hindlimb contralateral to the hindlimb in which inflammation had been induced with CFA 10 days prior to testing. Virally-expressed channelrhodopsin (ChR2) or halorhodopsin (NpHR) were used to modulate ACC activity. Extracellular recording tetrodes were also used to record ACC activity in response to noxious stimulation of the limbs. These techniques were combined with a conditioned place preference assay (CPA) to quantify the "aversive" responses from rats.

The paper begins by attempting to demonstrate that the paradigm induces "generalized enhancement of pain aversion". To do this the authors used a high power blue laser (50- to 250- mW) shined on the hindlimb to deliver a noxious heat stimulus to the hindpaw. Using three stimulus intensities they demonstrate that at the lowest intensity (50mW) mice do not develop a preference for either chamber of the PCA arena. In contrast, a 150mW (LS) or 250mW (HS) stimulus induces avoidance of the chamber in which the stimulus is applied. The authors show that the latency to a nocifensive response is significantly shorter at HS than LS. They interpret this result to mean that the HS stimulus is more noxious than the LS stimulus. While this might make sense intuitively, it is not consistent with what we know about spinally-mediated reflexes. Assuming that the spot size is consistent at LS and HS intensities (and it should be for this laser), the only difference to the rats should be the time it takes for the heat sensitive neuron to reach threshold. Once this threshold is reached, the spinal reflex is triggered and the movement should occlude further stimulation. Rats receiving the two different stimulus strengths should exhibit the same number of nocifensive behaviors (multiple presentations were made during the training period), the only difference being when during the stimulus application the behavior is triggered. If this is not the case (i.e. the 250mW stimulus is more noxious), the authors need to demonstrate this by showing greater activation of spinal circuits (e.g. via pERK stainining in dorsal horn neurons).

The authors build on this observation by injecting CFA into one hindpaw and 10 days later, testing rats in the PCA arena by applying the HS and LS stimuli to the hindlimb contralateral to the CFA-inflamed limb. Evidence that CFA has induced "generalized enhancement of pain aversion" is that the rats now spend even less time in the chamber associated with the LS stimulation (compared to pre-CFA), although they exhibit no change in response to be exposure to the HS stimulation. The authors conclude that the difference in response to the LS stimulus is due to an central mechanism (i.e., CNS) that has caused body-wide hypersensitivity. There are a number of problems with this conclusion. First, the difference in response to the LS stimulus, pre- and post- CFA is not very large and similar in magnitude to differences in responses time spent in the two chambers in the absence of any stimulus (e.g., Figure 1). Also making assessment of these results is difficult in that the number it is not clear whether the same rats were used for multiple experiments. Were the same rats tested over and over with the various stimuli, before and after CFA? Repeated use of the same rats will almost certainly complicate behavioral analysis. Finally, if the reason that there was no difference in the post-CFA response to the HS represented a ceiling affect, this would mean that whatever is regulating the "generalized enhancement of pain aversion" has a surprisingly limited range in being able to influence the response to painful sensations.

In the next experiment, tetrodes are used to record activity in the ACC (the proposed sited responsible for the "generalized enhancement of pain aversion") during hindpaw stimulation. Applying a population-decoding analysis using a support vector machine classifier the author's found that 53% of ACC neurons were tuned for the stimuli in that they exhibited greater firing for the HS vs. the LS. Given that these recordings are made in freely moving animals that may or may not have been previously stimulated in the CPA arena, it is probably not surprising that 53% of their neurons exhibited changes in firing frequency between the LS and HS. However, to conclude that all of these ACC neurons can code painful stimuli is difficult without more information. Following CFA-induced inflammation, there is no change in the percent of neurons identified as tuned (although the authors state there is a decrease (from 53% to 51%). The major difference identified by the authors was that there was a decrease in the accuracy of the decoding analysis with respect to its ability to detect differences between the LS and HS stimulus. This was attributed to the increase in response to the LS stimulus combined with no change in the response to the HS stimulus. The stability of the HS responses was attributed to a ceiling response to the stronger stimuli. As above, this interpretation is based on assumption that the HS stimulus caused more pain than the LS, for which there is no independent confirmation. Another issue raised by these results is the surprisingly high number of ACC neurons identified by the SVM as being "tuned" to pain intensity. Given all of the processes that have been attributed to the ACC it is surprising that half of the neurons would have this property. The major concern with this analysis is that in order to determine pain tuning, the authors used a 5 s window before and after application of the laser to determine which ACC neurons were part of pain circuit. However, the average response latency to the stimulus was ca. 1.5 sec for the HS and 3.0 for the LS stimulus. This means that for the two stimuli, different amounts of time following the pain reflex was used to determine the peak response (and calculate the Z-score). Thus, it is not surprising that the HS stimuli induced more activity in ACC neurons given there was more time for activation of CNS circuits. Had more time been allowed following the LS, it is possible that response would be equal between the two stimuli.

The final portion of the paper uses AAV to virally express either ChR2 or NpHR in the ACC and pairs activation of these opsins with hindlimb stimulation. No information is provided indicating the extent of expression in terms of number or types of neurons that express ChR2 or NpHR, other than a statement that pyramidal neurons were activated (a low power image is provided that is not particularly helpful other than indicating that the ACC was hit by the viral injection). ChR2 activation did not affect latency in response to stimulation of the hindpaw, but did increased aversion to LS, but not HS stimulation and occlude the difference between LS and HS in the CFA-treated rats. NpHR activation had the opposite affect.

Summary

The questions asked in this report are important and timely and the techniques employed are cutting edge. The way the bar graphs were presented was confusing and statistical analysis was hard to follow. More information is required on whether animals were used for multiple tests. The major problem concerned the assumption that the two stimuli produced different levels of pain and the sampling method used to compare the response to the two stimuli. Thus, the conclusion that the ACC is "necessary and sufficient" is premature. One would also like to see whether other areas that have been similarly implicated – insula, prefrontal cortex – had similar or different responses.

[Editors’ note: what now follows is the decision letter after the authors submitted for further consideration.]

Thank you for submitting your article "Chronic pain induces generalized enhancement of aversion" for consideration by *eLife*. Your article has been reviewed by three peer reviewers, one of whom is a member of our Board of Reviewing Editors, and the evaluation has been overseen by a Senior Editor. The reviewers have opted to remain anonymous.

The reviewers have discussed the reviews with one another and the Reviewing Editor has drafted this decision to help you prepare a revised submission.

Two of the reviewers of the original submission note that the new paper is improved, and that there is sufficient interest to merit publication in *eLife*. In order to assess the ACC electrophysiological and ontogenetic analysis, a new reviewer was invited to comment on the ACC physiology and technical aspects of the paper and its merits. This reviewer has raised serious concerns about the interpretation of the ACC physiological analysis and the absence of key controls for these experiments and certain statistical measures. A major concern the conclusion that ACC has a direct relationship to pain and pain responses. The electrophysiology section is problematic because the neurons are described as "pain-tuned". However, pain likely correlates strongly with movements of the mouse. Thus, similar results may be obtained with ephys in motor cortex. Therefore, whether the neurons exhibit "pain tuning" is considered premature due to lack of controls for movement. The bidirectional optogenetics experiments are more convincing than the ephys results. We would be happy to consider a revision that addresses the major concerns about interpretation of the ACC physiology and optogenetic manipulations as well as some of the statistical analyses.

Major points:

1) Electrophysiology is performed in ACC, and the authors are able to distinguish NS, LS, and HS conditions based on firing rates using an SVM classifier. They also show that manipulation of ACC can influence some of their behavioral measures of pain responses. My main issue with these results is that there are likely many behavioral variables that correlate with pain and that influence the behavioral readouts. Just because pain is what is considered here, it does not mean that is what ACC is encoding or influencing. Although there are many possible correlated variables, I will give an example using a single one: movement / motor efference copy. When the laser is applied to the forelimb, this will result in forelimb movement (withdrawal or smaller movements) and these movements might vary substantially between NS, LS, and HS conditions. It is well established that much of cortex (even sensory areas) receives motor efference copies. It therefore seems entirely possible that the spiking activity measured in Figure 2 could be related to movement and not pain. The authors have not done any controls to rule out movement. They would need to show that movement is the same between conditions or would need to show that movements outside a pain context do not trigger ACC activity. Relatedly, it is possible that activating or inactivating ACC causes changes in locomotor behavior. ACC is interconnected with motor regions and thus it might be possible that movement (or many other behavioral variables) could be perturbed rather than pain coding. Have the authors measured any features of locomotion in their experiments? Without at least ruling out movement cases, I am not convinced it is fair to conclude that ACC is encoding features directly related to pain. ACC could very well be encoding a different variable that just happens to covary with pain here.

2) The optogenetic experiments seem to need additional controls. First, there are no measurements of what the ChR2 and NpHR stimuli do to neural activity. It is assumed that they activate and inactive ACC, respectively. However, it seems important to show evidence that this is the case. It is dangerous to assume this just because of behavioral effects. For example, it is well established with microstimulation that inhibition can be rapidly recruited through synaptic connections and actually shut down excitatory activity. It seems essential to have some validation of the tools.

Also, it is common to do control experiments with laser light and a virus lacking the opsin. This controls for potential effects, like heating the brain or the visible light from the laser. For example, in Figure 3, the mice could be learning a paired association between the blue light (which is easy for them to see compared to the yellow light in Figure 4) and the pain from the laser to the forelimb (like in traditional fear conditioning). This pairing could drive their behavior in a more robust way than just the forelimb stimulus. Together these experiments seem important to verify that the effects are due to bidirectional modulation of ACC firing and not things like seeing the blue light.

3) The authors have quantified latency to withdrawal for the LS and HS stimuli. Have they also looked at the fraction of trials with a withdrawal?

4) In all experiments, how was the laser stimulus calibrated for the LS and HS stimuli? Was the latency to withdrawal for LS and HS similar for all experiments in Figure 1–Figure 4?

5) There are many places where statistics are missing. Statements are made about differences between figure panels but no statistics are provided. Some cases include:

– Comparing Figure 1 in Results paragraph 3

– Comparing Figure 1 in the same section

– Comparing Figure 2, in subsection “Chronic pain disrupts the ACC representation of acute pain signals, paragraph two”

– Comparing Figure 1,Figure 2, in subsection “Chronic pain impairs the bidirectional regulation of acute pain by the ACC”

– Comparing Figure 1,Figure 4, same section, paragraph three

– Comparing Figure 1,Figure 4, same section, paragraph three

6) The authors mention a change in slope between the plots in Figure 1. In the legend, the slopes are noted, but no statistics are provided to test if these slopes are significantly different.

7) I am not convinced by the occlusion results from Figure 3. Both CFA and ChR2 on their own cause a higher CPA score. This is due to a decrease in time spent in the conditioned chamber. With both of these cases, the time spent in the conditioned chamber approaches zero. When CFA and ChR2 are done together, there is no chance of ever seeing an additive effect because each one individually already approaches the floor (zero time in the conditioned chamber). Given that there is no chance of seeing an additive effect due to floor effects, this result is not meaningful. I suggest removing Figure 3.

8) For the SVM analyses, it would be good to show a chance level of decoding. For example, if the labels for the NS and HS trials are randomized in Figure 2 (for example with 1000 runs of different random assignments of labels), what are the bounds of the chance level of decoding achieved? Do these values fall outside the chance levels?

---

## [Author Response]

[Editors’ note: a previous version of this study was rejected after peer review, but the authors submitted for reconsideration. The first decision letter after peer review is shown below.]

*Reviewer #1:*

*This study reports that chronic pain caused by CFA injected into the paw causes a generalized enhancement in pain aversion, as measured by laser stimulation of the uninjured paw of the opposite limb. The authors make clever use of CPA to examine aversive values presented by acute pain signals, delivered by the laser. They found that a submaximal acute pain behavior elicited by a low-intensity noxious laser pulse (LS) is more affected by chronic pain than is maximal pain behavior elicited by high-intensity stimulus (HS). Presumably the HS is already maximally aversive. Physiologically, the neural representation of acute pain in the ACC elicited by the LS to the uninjured paw is altered in the chronic pain state. Although I AM not an expert in the decoding analysis used to conclude that chronic pain disrupts the ACC representation of LS, I find this study to have several interesting findings.*

*Concerns that would need to be addressed:*

*In the CPA measurement, the extent to which rats avoid the chamber in which LS is delivered in the CFA group is said to be increased compared to LS treatment of the uninjured rats (Figure 1). This observation forms the basis of the entire study, and yet I am not clear on the statistical test used to draw this conclusion. In general, the way in which CPA measurements were calculated and how statistical calculations and comparisons were made are confusing. This needs clarification.*

We sincerely apologize for the lack of detailed explanation for our experimental design and statistical analysis. In our CPA test (Figure 1), rats underwent three phases of experiments. During the first preconditioning (baseline) phase, they were left in the CPA chambers with free access, and the amount of time they spent in each chamber was measured to indicate baseline chamber preference. Next, rats were conditioned with a distinct stimulus in each treatment chamber for a total of 10 min. Thus, in Figure 1, during the conditioning phase, in one chamber rats received LS (low-intensity noxious stimulus), and in the other chamber they received NS (non-noxious stimulus). Finally, in the third “test” phase, conditioning stimuli were removed, and rats were allowed free access, and their movements were again recorded. We then calculated the time spent in each designated chamber at baseline (preconditioning) vs the test (postconditioning) phase. The treatment chambers were counterbalanced against treatment conditions to avoid innate preference. To present our data more clearly, we have reorganized the figures for the CPA data in our revised manuscript (Figure 1,Figure 3,Figure 4). In the new figures, the first two bars (white bars) represented the time spent in the designated chamber associated with each stimulus at baseline, and the next two bars (colored bars) showed the time spent for each chamber during the test phase. Thus, in Figure 1, at baseline, rats displayed no overt preference for either treatment chamber. During the test phase, however, naïve rats spent less time in the chamber paired with LS treatment, and more time in the chamber paired with NS. After a review of literature on the use of CPA in pain studies (Johansen et al., 2001; Johansen and Fields, 2004) and consultation with a statistician, Dr. Zhe Chen, a co-author in our study, in our revised manuscript we have used a paired Student’s t test to compare the time spent in each treatment chamber before and after conditioning (i.e. baseline vs test phase for each chamber). Our analysis yielded a statistically significant difference between baseline and test conditions for both LS and NS chambers, suggesting that rats spent significantly more time in the NS chamber and less time in the LS chamber during the test phase compared with baseline. We described this phenomenon as conditioned place aversion for the LS treatment. We then repeated this assay for rats after CFA injections to assess the conditioned place aversion for LS in the chronic pain condition, and the result is shown in Figure 1. Similarly we analyzed the rats’ ability to distinguish between HS (high-intensity noxious stimuli) and LS in the absence (Figure 1) or presence of chronic pain (Figure 1). We have expanded the Results section and modified our Figure 1 legend to provide a clear description of these data. Such method of data analysis was also applied to Figure 1 and Figure 4.

We also sincerely apologize for not providing a clear explanation in our previous manuscript on how we calculated the CPA scores or aversion scores for Figure 1. For such analysis, we adopted the methods from Fields and Porreca groups, who pioneered such studies (Johansen et al., 2001; Johansen and Fields, 2004; De Felice et al., 2013). For Figure 1, we analyzed the rats’ aversive response to LS when conditioned against NS. To do this, we subtracted the time rats spent in the LS chamber during the test phase from the time spent in that chamber at baseline. We called this difference the CPA score or aversion score for LS (when it is compared with NS). This CPA score indicates how much each rat avoided the LS chamber after conditioning with LS against NS, and this score gives us a quantitatively measure for how the rats could distinguish between LS and NS. This is how the Fields and Porreca groups computed their CPA scores (Johansen et al., 2001; Johansen and Fields, 2004; De Felice et al., 2013). We then compared this CPA score from saline-treated (control) rats with the CPA score from CFA-treated rats. This comparison is shown in Figure 1. We used an unpaired Student’s t testto calculate the statistical difference in the CPA scores between saline (control) and CFA treated rats. We found that CFA- treated rats (rats that experienced chronic pain) demonstrated a statistically significant increase in this CPA score, suggesting an increased avoidance of the LS chamber. Thus, rats in chronic pain demonstrate an increased aversion to a low-intensity noxious stimulus when compared with naïve rats. We then performed a similar analysis to compare the ability to distinguish between LS and HS stimuli in saline- vs CFA-treated rats (Figure 1). In this analysis, we subtracted the time spent in the HS chamber during the test phase from the time spent in that chamber at baseline.

We called this difference the CPA score for HS (when it is compared with LS). Effectively this CPA score reflects how much a rat avoids the HS chamber when it has to choose between HS and LS chambers after a short conditioning period. We then compared this CPA score for HS for saline-treated vs CFA-treated rats, using an unpaired Student’s t test. We found that CFA-treated rats, compared with saline-treated rats, demonstrated a decrease in the avoidance of the HS chamber as evidenced by a decreased CPA score. Thus, chronic pain causes a decrease in the rats’ ability to distinguish between the aversive values of HS and LS. As the reviewer astutely pointed out, Figure 1 formed an important behavioral basis for our study. These panels (Figure 1) demonstrate that rats after chronic pain show an increased ability to distinguish between non-noxious and low-intensity noxious signals, and a decreased ability to distinguish between low-intensity and high-intensity noxious signals. Thus, there is a disturbance in pain scale in these rats, likely indicating that rats in chronic pain interpret even low-intensity stimulus as highly noxious. These results are remarkably compatible with results from clinical studies. We have used similar methods to calculate CPA scores under slightly different experimental conditions in Figure 3,Figure 4 as well.

Again, we would like to sincerely apologize for the lack of clarity in explaining our behavior data. We have modified the text in the Results section and figure legends to provide greater clarity for our use of the CPA score. We have also expanded the Results and Materials and methods section to provide a clearer explanation of our study methods and statistical analysis (see revised Materials and methods section, particularly the expanded Statistical Analysis subsection).

*Whether alterations of central representation of LS in the chronic pain state is unique to the ACC or whether the ACC is one component of an interconnected network of brain regions similarly altered is not clear. That is, does optogenetic perturbation of any component of an interconnected network (medial thalamus, etc) lead to similar findings? Is the ACC the locus of altered neural representation of acute pain to an uninjured limb or one component of many?*

We appreciate this astute comment from the reviewer. We did not perform experiments on other brain regions in this study. We believe that the present finding of ACC involvement is of sufficient interest to the pain research community, as we are the first group to identify this generalized enhancement of aversion and a potential neural substrate for this behavior. We also hypothesize that the ACC is likely one crucial component in an interconnected network of brain regions that fully regulate this important behavior in the chronic pain state. We have included such discussion in our Discussion section (second to last paragraph of the new manuscript). We also plan to perform future studies to investigate the role of other cortical regions such as the prelimbic prefrontal cortex in this behavioral condition, but such studies are beyond the scope of our current manuscript.

*Reviewer #2:*

[…]

*Major concerns.*

*1) I feel that the descriptions of how the data were analyzed are insufficient for me to determine whether the analyses were done correctly. For instance, many graphs show the time spent on each side of the chamber before and after conditioning treatment. In each case, the data presented in the first two columns are the mirror image of the data presented in the second two columns (since the rats are either in one chamber or the other, so this is not an independent variable.) To analyze this type of data, the authors do a 2-way ANOVA, which I find confusing. In other experiments, they calculate a CPA score, which is defined as the difference between the time spent in either treatment chamber. Again, I'm not exactly sure which numbers were subtracted to give the resulting data. I think these experiments would benefit from a clearer description as well as careful analysis by a statistical expert (not me). The authors should also clarify whether these experiments were done on the same cohort of rats, or whether naive rats were used for each experiment.*

We sincerely apologize for the lack of detailed explanation for our experimental design and statistical analysis. In our CPA test (Figure 1), rats underwent three phases of experiments. During the first preconditioning (baseline) phase, they were left in the CPA chambers with free access, and the amount of time they spent in each chamber was measured to indicate baseline chamber preference. Next, rats were conditioned with a distinct stimulus in each treatment chamber for a total of 10 min. Thus, in Figure 1, during the conditioning phase, in one chamber rats received LS (low-intensity noxious stimulus), and in the other chamber they received NS (non-noxious stimulus). Finally, in the third “test” phase, conditioning stimuli were removed, and rats were allowed free access, and their movements were again recorded. We then calculated the time spent in each designated chamber at baseline (preconditioning) vs the test (postconditioning) phase. The treatment chambers were counterbalanced against treatment conditions to avoid innate preference. To present our data more clearly, we have reorganized the figures for the CPA data in our revised manuscript (Figure 1,Figure 3,Figure 4). In the new figures, the first two bars (white bars) represented the time spent in the designated chamber associated with each stimulus at baseline, and the next two bars (colored bars) showed the time spent for each chamber during the test phase. Thus, in Figure 1, at baseline, rats displayed no overt preference for either treatment chamber. During the test phase, however, naïve rats spent less time in the chamber paired with LS treatment, and more time in the chamber paired with NS.

After a review of literature on the use of CPA in pain studies (Johansen et al., 2001; Johansen and Fields, 2004) and consultation with a statistician, Dr. Zhe Chen, a co-author in our study, in our revised manuscript we have used a paired Student’s t test to compare the time spent in each treatment chamber before and after conditioning (i.e. baseline vs test phase for each chamber). Thus, in Figure 1, for example, our analysis yielded a statistically significant difference between baseline and test conditions for both LS and NS chambers, suggesting that rats spent significantly more time in the NS chamber and less time in the LS chamber during the test phase compared with baseline. We described this phenomenon as conditioned place aversion for the LS treatment. We then repeated this assay for rats after CFA injections to assess the conditioned place aversion for LS in the chronic pain condition, and the result is shown in Figure 1. Similarly, we analyzed the rats’ ability to distinguish between HS (high-intensity noxious stimuli) and LS in the absence (Figure 1) or presence of chronic pain (Figure 1). We have expanded the Results section and modified our Figure 1 legend to provide a clear description of these data. Such method of data analysis was also applied to Figure 1 and 4.

We also sincerely apologize for not providing a clear explanation in our previous manuscript on how we calculated the CPA scores or aversion scores for Figure 1. For such analysis, we adopted the methods from Fields and Porreca groups, who pioneered such studies (Johansen et al., 2001; Johansen and Fields, 2004; De Felice et al., 2013). For example, in Figure 1, we analyzed the rats’ aversive response to LS when conditioned against NS. To do this, we subtracted the time rats spent in the LS chamber during the test phase from the time they spent in that chamber at baseline. We called this difference the CPA score or aversion score for LS (when it is compared with NS). This CPA score indicates how much each rat avoided the LS chamber after conditioning with LS against NS, and this score gives us a quantitatively measure for how the rats could distinguish between LS and NS. This is how the Fields and Porreca groups computed their CPA scores (Johansen et al., 2001; Johansen and Fields, 2004; De Felice et al., 2013). We then compared this CPA score from saline-treated (control) rats with the CPA score from CFA-treated rats. This comparison is shown in Figure 1. We used an unpaired Student’s t test to calculate the statistical difference in the CPA scores between saline (control) and CFA treated rats. We found that CFA-treated rats (rats that experienced chronic pain) demonstrated a statistically significant increase in this CPA score, suggesting an increased avoidance of the LS chamber. Thus, rats in chronic pain demonstrate an increased aversion to a low-intensity noxious stimulus when compared with naïve rats. We then performed a similar analysis to compare the ability to distinguish between LS and HS stimuli in saline- vs CFA-treated rats (Figure 1). In this analysis, we subtracted the time spent in the HS chamber during the test phase from the time spent in that chamber at baseline. We called this difference the CPA score for HS (when it is compared with LS). Effectively this CPA score reflects how much a rat avoids the HS chamber when it has to choose between HS and LS chambers after a short conditioning period. We then compared this CPA score for HS for saline-treated vs CFA-treated rats, using an unpaired Student’s t test. We found that CFA-treated rats, compared with saline-treated rats, demonstrated a decrease in the avoidance of the HS chamber as evidenced by a decreased CPA score. Thus, chronic pain causes a decrease in the rats’ ability to distinguish between the aversive values of HS and LS. We have used similar methods to calculate CPA scores under slightly different experimental conditions in Figure 3 and Figure 4.

Again, we would like to sincerely apologize for the lack of clarity in explaining our behavior data. We have modified the text in the Results section and figure legends to provide greater clarity for our use of the CPA score. We have also expanded the Results and Materials and methods section to provide a clearer explanation of our study methods and statistical analysis (see revised Materials and methods section, particularly the expanded Statistical Analysis subsection).

Finally, in terms of the cohorts of animals, some (but by no means all) of the cohorts were used for more than one behavior test, such as the tests between LS vs NS, LS vs HS and HS vs NS. The purpose for using the same rats for these three tests is to confirm that the same rat could demonstrate the ability to differentiate between LS and NS, LS and HS and HS and NS. The confirmation of this behavioral consistency is important for our study, as we seek to establish that rats could distinguish between different pain intensities. However, in those cases of repeated testing, the following measures were taken to avoid behavioral sensitization. 1) We avoided testing in any particular order. Thus, different rats underwent different behavior tests at different time points (some rats underwent LS vs NS first, whereas other rats underwent LS vs HS first, etc). 2) A period of several days was designated for rest between different test conditions. 3) Stimulations were paired with different chambers during different assays for the same rat so as to avoid chamber association over time. As the results of the above measures, we did not observe any behavioral sensitization in our assays. We also did not observe any long lasting aversive memory, as rats returned to baseline non-preference for any chamber at the start of CPA on any given day. It should be noted that to further ensure the overall reproducibility of our data, we have taken the following additional measures. 1) Multiple cohorts were used for each behavior test, and results as shown in each behavior data panel in Figure 1,Figure 3,Figure 4 come from 2-4 cohorts of rats. For additional safeguard against any possible interference on behavior tests, different cohorts underwent testing in different orders. 2) We have used saline controls whenever appropriate (Figure 1,Figure 3). 3) Finally, in order to confirm the reproducibility of our data, we have repeated each of the CPA tests again with new rats, increasing the n for each experiment by approximately 25%. All additional data points have been included in the revised submission.

*2) Conceptually, I find it hard to wrap my head around the difference between the low intensity noxious stimulus and the high intensity noxious stimulus. In each case, the rat is exposed to a laser until (and only until) the stimulus triggers a nociceptive reflex. Granted, the authors show that in CPA tests the mice can distinguish LS from HS. And they also show that a declassifying code can distinguish responses from LS and HS. Nevertheless, I feel that, in the spectrum of possible pain, the two types of stimuli would be rather similar. For this reason, I find it peculiar that the optogenetic stimulation of the ACC affects behavioral responses to LS but not HS. The reason given "HS presumably already elicited maximal ACC response, resulting in a less pronounced effect of optogenetic activation" is both unsatisfying and overinterpreted (note that there is actually no effect of optogenetic stimulation in the HS condition).*

We appreciate this thoughtful comment from the reviewer. In fact, there are several lines of evidence that support a significant difference in the noxious intensity between LS and HS. First, LS and HS are driven by different power outputs from our blue laser. LS corresponds to 150mW of power output, whereas HS corresponds to 250mW. In response to the reviewer, we measured the temperature generated by LS and HS using a temperature sensor. The temperature generated by LS after 3s (average time of paw withdrawal) was 53.9+2.1 ^o^C, whereas the temperature generated by HS after 1.5s (time of paw withdrawal) was 61.42+1.6 ^o^C. We have added this data as Figure 1—figure supplement 1. The reason that HS achieved higher temperature than LS at the time of paw withdrawal is that heat transfer from the laser to the tissue happens on a nonlinear time scale. This transfer occurs faster than the physical movement of paw withdrawal, is faster for HS than for LS due to the higher power output of HS, and can persist even after the immediate removal of the direct heat source (laser). This, in fact, is not surprising. Human studies have not equated withdrawal responses with pain intensity. In most human studies, even though subjects withdraw in response to a variety of noxious stimuli, the actual pain intensity experienced with each stimulus is different and graded by self-report. In this study we try to show that we can use CPA as a similar test to assess pain intensity in rodents. Secondly, unlike a CO2 laser which has focused tissue penetration, the laser we use emits visible light and has a diffuse tissue penetration. Hence, in addition to an increase in peak temperature, an increase in laser output power can also cause increased depth and area of tissue penetration. Thus, HS effectively activated more TRP channels on more nociceptive afferent neurons than LS. A third line of evidence comes from our behavior data. As pointed out by the reviewer, both the latency to withdrawal and CPA results indicate that naïve rats could distinguish between LS and HS (Figure 1). Finally, as the reviewer pointed out, our machine learning decoding analysis, which was unbiased and independent from behavioral phenotypes, was also able to decode the difference between HS and LS based on neural data.

In terms of the optogenetic effect on LS and HS, our data indicates that stimulation of the ACC increased the aversive response to LS in a way that is statistically significant (Figure 3).

After we have repeated our experiments on additional rats, we found that stimulation of the ACC can indeed increase the aversive response to HS, but such a response is not statistically significant (Figure 3). There are two possible interpretations for such findings. First, at the physiological level, clinical postoperative studies have shown that the pain scale (0-10) can be disrupted in chronic pain patients, but the maximal pain score does not always change. We believe that our CPA test serves as a similar pain scale report for rats, and thus with very highly noxious stimulation (HS), there could be a maximal aversive response expressed by the rats in our experiments, similar to a maximal pain score experienced by patients. A secondpossible interpretation for this data is that there may be a limit on the aversive response that could be tested with our CPA assay. Thus, it is the ceiling effect of the assay, rather than native physiology, which was responsible for these results. As we could not distinguish between these two possibilities, we agree with the reviewer that our statement of “maximal ACC response” would be an over-interpretation. We have revised our interpretation in the Results section of our manuscript accordingly. Please note, however, that inhibition of the ACC does reduce the aversive effect of HS (Figure 4). Thus, the aversive response to HS on our CPA assay is modifiable uni-directionally by alterations of ACC functions. We would like to emphasize, in addition, that these results do not affect the central thesis of our study, which is that aversive response to low-intensity stimulation is elevated by the presence of chronic pain and that the ACC plays an important role in such an altered aversive response.

*3. It seems to me that a key control is the effect of optogenetic inhibiton of the ACC on CPA by itself (or paired with NS). This control is missing.*

We appreciate this astute comment and have performed this control experiment. The results are shown in Figure 4—figure supplement 2, and they demonstrate that inhibition of the ACC does not have any effect on the CPA when paired with NS.

*Reviewer #3:*

[…]

*The paper begins by attempting to demonstrate that the paradigm induces "generalized enhancement of pain aversion". To do this the authors used a high power blue laser (50- to 250- mW) shined on the hindlimb to deliver a noxious heat stimulus to the hindpaw. Using three stimulus intensities they demonstrate that at the lowest intensity (50mW) mice do not develop a preference for either chamber of the PCA arena. In contrast, a 150mW (LS) or 250mW (HS) stimulus induces avoidance of the chamber in which the stimulus is applied. The authors show that the latency to a nocifensive response is significantly shorter at HS than LS. They interpret this result to mean that the HS stimulus is more noxious than the LS stimulus. While this might make sense intuitively, it is not consistent with what we know about spinally-mediated reflexes. Assuming that the spot size is consistent at LS and HS intensities (and it should be for this laser), the only difference to the rats should be the time it takes for the heat sensitive neuron to reach threshold. Once this threshold is reached, the spinal reflex is triggered and the movement should occlude further stimulation. Rats receiving the two different stimulus strengths should exhibit the same number of nocifensive behaviors (multiple presentations were made during the training period), the only difference being when during the stimulus application the behavior is triggered. If this is not the case (i.e. the 250mW stimulus is more noxious), the authors need to demonstrate this by showing greater activation of spinal circuits (e.g. via pERK stainining in dorsal horn neurons).*

We appreciate this careful analysis from the reviewer. However, we believe that there are several lines of evidence that support a difference in the noxious intensity between LS and HS. First, in terms of the physical property of the heat stimuli, LS corresponds to 150mW of power output from the blue laser that we use, whereas HS corresponds to 250mW. In response to the reviewer, we measured the temperature generated by LS and HS using a temperature sensor. The temperature generated by LS after 3s (average time of paw withdrawal) was 53.9+2.1 ^o^C, whereas the temperature generated by HS after 1.5s (time of paw withdrawal) was considerably higher at 61.42+1.6 ^o^C. We have added this data as an independent demonstration of the greater noxious intensity of HS in Figure 1—figure supplement 1. It is important to note that a typical laser generates heat on a nonlinear time scale. The energy generated by the laser (E) = W*t, where W is laser power and t is time. If the energy can be absorbed totally by skin, Q (heat) =E. Due to the higher power output, HS can achieve a steeper increase in temperature over time compared with LS. According to the heat transfer formula Q=mCΔT, the change of temperature ΔT depends on the skin specific heat capacity (C) and mass (m) that is being heated. Thus, ΔT=Q/mC=Wt/mC. It should be noted that m and C are dependent on the properties of the rat skin, and these values may be slightly different from those derived from the temperature sensor. Importantly, the temperature in the rat paws after laser stimulation increases in a non-linear fashion, and heat transfer from the laser to tissue happens on a faster scale than the physical movement of withdrawal, and faster for HS than for LS. Hence HS can achieve a higher peak temperature than LS. This is the same case with human studies which use lasers to trigger pain. In human studies, even though subjects withdraw in response to a variety of noxious stimuli, the actual pain intensity experienced with each stimulus is different and has to be graded by self- report. In this study we try to show that we can use CPA as a similar report to assess pain intensity in rodents.

Secondly, unlike a CO2 laser which has focused tissue penetration, the blue laser we used emits visible light and has a diffuse tissue penetration. We apologize if we did not make this point clear. Thus, an increase in laser power causes not only an increase in temperature at a focal point (as measured by our temperature sensor) but also an increased depth and area of tissue penetration. As a result, HS effectively activated more TRP channels on more nociceptive afferent neurons than LS.

A thirdline of evidence comes from our behavior data. Both the latency to withdrawal and CPA results indicate that naïve rats could distinguish between LS and HS very well (Figure 1). It is particularly important to note that naïve rats withdrew their paws to both LS and HS, as the reviewer pointed out, and yet they still avoided the HS chamber when it was compared with LS during conditioning on the CPA test, strongly indicating that HS is more aversive than LS (Figure 1).

Finally, our machine learning decoding analysis, which is unbiased and independent from behavioral results, is also able to detect the difference between HS and LS based on the firing rates of ACC neurons.

Based on these four lines of evidence, we believe that HS provides a greater noxious value than LS.

With regards to the experiment suggested by the reviewer, we think that it is a very interesting idea. Phosphorylation of ERK is important for synaptic plasticity and hence a key step in central sensitization. In a landmark paper, staining of pERK in dorsal horn neurons was used to demonstrate that “nociceptive-specific activation of ERK in spinal neurons contributes to pain hypersensitivity” (Ji et al., 1999). Subsequent papers have also show ERK to be a key link between peripheral nociceptive inputs and long lasting spinal activation (Kawasaki et al., 2004). However, there are key differences between our current study and these important studies. First, thematically, our study intends to show that chronic pain alters the aversive valuation of transient acute pain signals. In contrast to the studies cited above, the acute aversive responses elicited by LS and HS are meant to be temporary, and they were not meant to result from spinal sensitization. Indeed, we did not observe any behavioral or electrophysiological evidence for sensitization. In our study, we did not observe a decrease in paw withdrawal latency after LS or HS during the CPA or electrophysiological recordings. In addition, our electrophysiological data did not demonstrate time-dependent increases in spike rate responses. Second, at the technical level, in most of the studies examining the role of pERK in spinal sensitization, the peripheral stimulation was either intense (direct electrical stimulation of c-fiber) or had lasting qualities such as nerve injury or persistent inflammation. However, the stimulations in our study were transient and did not last more than 3 seconds in the case of LS or 1.5 seconds in the case of HS. Any stimulus was also followed by 10 seconds of no stimulation, and the whole conditioning phase lasted 10 minutes. This means a total of (600/(10+3)x3=138 seconds of LS stimulation or (600/(10+1.5)x1.5=78 seconds of HS stimulation. These calculations suggest that HS and LS are not likely to cause persistent spinal activation, further supporting our idea that the ACC can regulate transient or acute aversive signals, not just chronic pain. Therefore, we believe that in our study, neither LS nor HS was sufficient to trigger phosphorylation of ERK in dorsal horn neurons, and hence the proposed experiment is not likely to work. However, we feel that the four reasons above provided sufficient rationale for why HS was more noxious than LS.

*The authors build on this observation by injecting CFA into one hindpaw and 10 days later, testing rats in the PCA arena by applying the HS and LS stimuli to the hindlimb contralateral to the CFA-inflamed limb. Evidence that CFA has induced "generalized enhancement of pain aversion" is that the rats now spend even less time in the chamber associated with the LS stimulation (compared to pre-CFA), although they exhibit no change in response to be exposure to the HS stimulation. The authors conclude that the difference in response to the LS stimulus is due to an central mechanism (i.e., CNS) that has caused body-wide hypersensitivity. There are a number of problems with this conclusion. First, the difference in response to the LS stimulus, pre- and post- CFA is not very large and similar in magnitude to differences in responses time spent in the two chambers in the absence of any stimulus (e.g., Figure 1). Also making assessment of these results is difficult in that the number it is not clear whether the same rats were used for multiple experiments. Were the same rats tested over and over with the various stimuli, before and after CFA? Repeated use of the same rats will almost certainly complicate behavioral analysis. Finally, if the reason that there was no difference in the post-CFA response to the HS represented a ceiling affect, this would mean that whatever is regulating the "generalized enhancement of pain aversion" has a surprisingly limited range in being able to influence the response to painful sensations.*

We appreciate these careful observations from the reviewer. To address the reviewer’s firstconcern, we have reorganized our data presentation in the new manuscript. This organization allows a better visualization for our CPA data (Figure 1,Figure 3,Figure 4). In the new panels, the first two bars represented the time spent in the designated chamber associated with each stimulus at baseline (prior to conditioning), and the next two bars showed the time spent for each chamber during the test phase (after conditioning). Thus, in Figure 1, at baseline, rats displayed no overt preference for either treatment chamber. During the test phase, however, naïve rats spent less time in the chamber paired with LS treatment, and more time in the chamber paired with NS. Comparing Figure 1with Figure 1, however, it is clear that CFA-treated rats displayed a significant increase in the avoidance of LS chamber. In order to more rigorously analyze the difference between LS and NS or LS and HS, we have also compared rats that received saline injection with CFA-treated rats.

These data are shown in Figure 1 and J. Figure 1 demonstrate the difference in the aversive response to the LS stimulus between saline-treated and CFA-treated rats. To make such comparisons, we adopted the methods developed by the Fields and Porreca groups to calculate a CPA score or aversion score (Johansen et al., 2001; Johansen and Fields, 2004; De Felice et al., 2013). For Figure 1, we analyzed the rats’ aversive response to LS when they were conditioned with LS and NS. To do this, we calculated a CPA score by subtracting the time spent in the LS chamber during the test phase from the time spent in that chamber at baseline. This CPA score indicates how much each rat avoided the LS chamber after conditioning with LS against the NS stimulus, and it gives us a quantitatively measure for how the rats could distinguish between LS and NS. We then compared CPA scores from saline- treated (control) rats with CPA scores from CFA-treated rats. This comparison is shown in Figure 1. We used an unpaired Student’s t test to calculate the statistical difference between control and CFA-treated rats and found that CFA rats (rats that experienced chronic pain) demonstrated a statistically significant increase in the CPA scores, suggesting an increased avoidance of the LS chamber compared with the NS chamber. Thus, rats in chronic pain demonstrate an increased aversion to a low-intensity noxious stimulus when compared with naïve rats. We then performed a similar analysis to compare the ability to distinguish between LS and HS stimuli in saline- vs CFA-treated rats (Figure 1). In this analysis, we calculated a different CPA score by subtracting the time spent in the HS chamber during the test phase from the time spent in that chamber at baseline. This CPA score quantifies how much a rat avoids the HS chamber when it is conditioned with HS and LS. We then compared these CPA scores for saline-treated vs CFA- treated rats, using an unpaired Student’s t test. We found that CFA-treated rats, compared with saline-treated rats, demonstrated a decrease in the avoidance of the HS chamber when conditioned against LS. Thus, chronic pain causes a decrease in the rats’ ability to distinguish between the aversive values of HS and LS. Figure 1 formed the important behavioral basis for our study. These panels (Figure 1) demonstrate that rats after chronic pain show an increased ability to distinguish between non-noxious and low-intensity noxious signals, and a decreased ability to distinguish between low-intensity and high-intensity noxious signals. Thus, there is a disturbance in pain scale in these rats, indicating that rats in chronic pain interpret even low-intensity stimulus as highly noxious. These results are remarkably compatible with results from clinical studies (Scudds et al., 1987; Petzke et al., 2003; Kehlet et al., 2006; Kudel et al., 2007; Scott et al., 2010). We sincerely apologize for the lack of clarity in explaining our behavior data. We have expanded the Results and Materials and methods section to provide a clear explanation of our study methods and statistical analysis.

To address the reviewer’s secondconcern, in terms of the cohorts of animals, some (but by no means all) of the cohorts were used for more than one behavior test, such as the tests between LS vs NS, LS vs HS or HS vs NS. The purpose for using the same rats for these three tests is to confirm that the same rat could demonstrate the ability to differentiate between LS and NS, LS and HS and HS and NS. The confirmation of this behavioral consistency is important for our study, as we seek to establish that rats could distinguish between different pain intensities.

However, in those cases of repeated testing, the following measures were taken to avoid behavioral sensitization. 1) We avoided testing in any particular order. Thus, different rats underwent different behavior tests at different time points (some rats underwent LS vs NS first, whereas other rats underwent LS vs HS first, etc). 2) A period of several days was designated for rest between different test conditions. 3) Stimulations were paired with different chambers during different assays for the same rat so as to avoid chamber association over time. As the results of the above measures, we did not observe any behavioral sensitization in our assays. We also did not observe any long lasting aversive memory, as rats returned to baseline non- preference for any chamber at the start of CPA on any given day. It should be noted that to further ensure the overall reproducibility of our data, we have taken the following additional measures. 1) Multiple cohorts were used for each behavior test, and results as shown in each behavior data panel in Figure 1,Figure 3,Figure 4 come from 2-4 cohorts of rats. For additional safeguard against any possible interference on behavior tests, different cohorts underwent testing in different orders. 2) We have used saline controls whenever appropriate (Figure 1,Figure 3). 3) Finally, in order to confirm the reproducibility of our data, we have repeated each of the CPA tests again with new rats, increasing the n for each experiment by approximately 25%. All additional data points have been included in the revised submission.

Finally, in terms of the ceiling effect of the aversive response to noxious stimuli, we feel that such findings are not surprising. There are two possible interpretations for such findings. First, at the physiological level, clinical postoperative studies have shown that the pain scale (0-10) can be disrupted in chronic pain patients, but the maximal pain score does not always change. We believe that our CPA test serves as a similar pain scale report for rats, and thus with very highly noxious stimulation (HS), there could be a maximal aversive response expressed by the rats in our experiments, similar to a maximal pain score experienced by patients. A secondpossible interpretation for this data is that there may be a limit on the aversive response that could be tested with our CPA assay. Thus, it is the ceiling effect of the assay, rather than native physiology, which was responsible for these results. We have provided this explanation in the Results sections of our revised manuscript. Please note, however, that such ceiling effects do not affect the central thesis of our study, which is that aversive response to low-intensity stimulation is elevated by the presence of chronic pain and that the ACC plays a role in such an altered aversive response.

*In the next experiment, tetrodes are used to record activity in the ACC (the proposed sited responsible for the "generalized enhancement of pain aversion") during hindpaw stimulation. Applying a population-decoding analysis using a support vector machine classifier the author's found that 53% of ACC neurons were tuned for the stimuli in that they exhibited greater firing for the HS vs. the LS. Given that these recordings are made in freely moving animals that may or may not have been previously stimulated in the CPA arena, it is probably not surprising that 53% of their neurons exhibited changes in firing frequency between the LS and HS. However, to conclude that all of these ACC neurons can code painful stimuli is difficult without more information.*

We apologize for the lack of clear explanation for our encoding and decoding analyses. As the reviewer astutely pointed out, it is not surprising that >50% of ACC neurons which responded to pain signals showed changes in firing frequency. After consultation with Dr. Zhe Chen, a co-author on this manuscript who is a statistician and computational neuroscientist, we have decided to use a very stringent set of criteria to define pain responsive neurons. These criteria are now carefully explained in the (Statistical Analysis subsection of) Materials and methods section of our revised manuscript. Based on these criteria, fewer neurons could be categorized as pain responsive. In addition, pain tuning neurons were identified as those pain responsive neurons which further displayed increased firing rate in response to HS compared with LS. For the sake of clarification, we have revised Figure 2, including the pie charts in Figure 2, and we have provided additional specific information in the Results section and Figure 2 legend regarding the fraction of pain tuning neurons.

Secondly, as indicated in the Materials and methods section of our manuscript, a support vector machine analysis was used to analyze a group of neurons during a recording session comprised of 60 trials. Importantly, the contribution of individual neurons to decoding was weighed during the training trials. Some neurons provided high degrees of coding information; others did not.

Those neurons that did not report high coding efficacy were assigned smaller weights in our support vector machine algorithm, whereas neurons that displayed higher coding efficacy had higher weights. We did not a prioriassign the weights, but our algorithm assigned the weights based on actual data during the training trials. In other words, our machine learning algorithm “learned” that individual neurons could encode varying degrees of information regarding pain intensity and took this information into consideration during the test runs. The purpose of this decoding analysis is to provide unbiased assessment of the importance of ACC neurons in decoding pain intensity, and so naturally it is vital that we used all the neurons in our decoding algorithm. As our results show (Figure 2), our algorithm analyzed all the neurons in a test session and used this composite information to predict if the stimulus was HS or LS with a high degree of accuracy in naïve rats. In agreement with the reviewer, we cannot conclude, based on such analysis, that all the neurons in the ACC are important for pain decoding. What we can say, however, is that pain tuning and non-tuning neurons together provide enough information for the computer to detect pain intensity. In fact, this demonstrates the power of machine learning analysis: it is able to achieve very accurate pain decoding even when the neurons provide highly heterogeneous information (with a large number of neurons responding very little to noxious stimuli). We apologize for the lack of explanation of our decoding methods, and we have expanded the decoding section in our Results section to explain our approach and our data more clearly.

*Following CFA-induced inflammation, there is no change in the percent of neurons identified as tuned (although the authors state there is a decrease (from 53% to 51%). The major difference identified by the authors was that there was a decrease in the accuracy of the decoding analysis with respect to its ability to detect differences between the LS and HS stimulus. This was attributed to the increase in response to the LS stimulus combined with no change in the response to the HS stimulus. The stability of the HS responses was attributed to a ceiling response to the stronger stimuli. As above, this interpretation is based on assumption that the HS stimulus caused more pain than the LS, for which there is no independent confirmation.*

We appreciate this comment from the reviewer. There are several lines of evidence that support a significant difference in the noxious intensity between LS and HS. First, LS and HS are driven by different power outputs from our laser. LS corresponds to 150mW of power output, whereas HS corresponds to 250mW. The temperature generated by LS after 3s (time of paw withdrawal) is approximately 53.9+2.1 ^o^C by measurement from a temperature sensor, whereas the temperature generated by HS after 1.5s (time of paw withdrawal) is approximately 61.42+1.6 ^o^C. We have added this data as an independentdemonstration of the greater noxious intensity of HS in Figure 1—figure supplement 1. In addition, heat transfer from the laser to the tissue happens on a nonlinear time scale, and it can happen faster than the actual movement of paw withdrawal. Thus, despite the common time stamp of paw withdrawal, HS achieved a substantially higher peak temperature than LS in our study. Secondly, unlike a CO2 laser which has focused tissue penetration, the laser we use emits visible light and has a diffuse tissue penetration. Hence, in addition to an increase in peak temperature, an increase in laser output power can also cause increased depth and area of tissue penetration. Thus, HS effectively activated more TRP channels on more nociceptive afferent neurons than LS. A thirdline of evidence comes from our behavior data. Both our latency to withdrawal and CPA results indicate that naïve rats could distinguish between LS and HS (Figure 1).

*Another issue raised by these results is the surprisingly high number of ACC neurons identified by the SVM as being "tuned" to pain intensity. Given all of the processes that have been attributed to the ACC it is surprising that half of the neurons would have this property.*

We apologize for the lack of clear explanation for our methods to identify pain responsive and tuning neurons. In our new revised manuscript, we used a very stringent set of criteria for identifying neurons that responded to pain. These criteria are now carefully explained in the (Statistical Analysis subsection of) Materials and methods section of our manuscript. Based on these criteria, few neurons could be categorized as pain responsive. In addition, pain tuning neurons were identified as those pain responsive neurons which displayed increased firing rate in response to HS compared with LS. There are approximately 15% of neurons in the ACC which are pain-tuning by our criteria. To provide further clarification, we have revised our Figure 2, including the pie charts in Figure 2 to clearly indicate the fraction of neurons that were pain responsive and at the same time displayed tuning properties. Meanwhile, we would like to emphasize that SVM was notused to identify pain responsive or tuning neurons. SVM was used to decode whether a random sample of ACC neurons, which contains a combination of pain responsive and non-responsive neurons, could predict the noxious intensity. Due to the unbiased nature of this analysis, we did not tell the algorithm which neurons were pain-tuning, and which ones were not. Our SVM algorithm assigned a weight to each individual neuron, depending on the relevance of the firing rate changes in that neuron in response to a noxious stimulus (please refer to the Materials and methods section for details on how this weight was calculated). We then used spike information from all neurons in a test session to decode noxious intensity. Thus, the goal of our SVM analysis is to demonstrate an unbiased assessment for the capability of neurons in the ACC for pain-intensity decoding. The purpose for such analysis is to provide another way to confirm that an ensemble of ACC neurons carries meaningful information regarding the aversive value of noxious stimulation. We have revised our Results section to make this point more clear.

*The major concern with this analysis is that in order to determine pain tuning, the authors used a 5 s window before and after application of the laser to determine which ACC neurons were part of pain circuit. However, the average response latency to the stimulus was ca. 1.5 sec for the HS and 3.0 for the LS stimulus. This means that for the two stimuli, different amounts of time following the pain reflex was used to determine the peak response (and calculate the Z-score). Thus, it is not surprising that the HS stimuli induced more activity in ACC neurons given there was more time for activation of CNS circuits. Had more time been allowed following the LS, it is possible that response would be equal between the two stimuli.*

We appreciate this comment from the reviewer. In our encoding analysis, in order to define a pain responsive neuron, we used 5s window before and after laser stimulation. While there is a difference in withdrawal latency to LS and HS, we purposely used a long enough time window to minimize any potential bias. The calculated z-score is based on baseline recording prior to stimulus and thus should not be affected by the timing of withdrawals. After stimulus, neural responses typically occurred within 3 seconds (similar to the timing of withdrawals but in many cases earlier than withdrawals), and we did not observe multiple peaks of spike rate increases. Thus when we used peak z-score over 5 seconds after stimulation to identify pain responsive neurons, we did not bias in favor of HS. To confirm the impartiality of our approach, we have also performed unbiased decoding analysis using a shorter time window (3 seconds) after stimulus. We have shown this data in Figure 2—figure supplement 1 of our revised manuscript. As this figure shows, in fact, a 3 second time window is sufficient to provide decoding with similar accuracy as 5 seconds. This additional result provided confirmation that most of the neural changes peaked well within a 5 second period after peripheral stimulation, and that based on such information we could reliably predict the intensity of stimulation.

The final portion of the paper uses AAV to virally express either ChR2 or NpHR in the ACC and pairs activation of these opsins with hindlimb stimulation. No information is provided indicating the extent of expression in terms of number or types of neurons that express ChR2 or NpHR, other than a statement that pyramidal neurons were.

Since we used a CAMKII promotor, we expect that excitatory pyramidal neurons are most likely infected. We have done additional staining experiments to verify this finding, using VGLUTs as excitatory neuronal marker and DAPI as cellular marker. These control experiments are shown in Figure 3—figure supplement 1 and Figure 4—figure supplement 1. As can be seen in these figures, we achieved high expression of opsins in excitatory neurons. We have also assessed the efficacy of opsin expression using these markers. Up to 90% of YFP positive neurons (for both ChR2 and NpHR constructs) stained positively for VGLUTs.

[Editors’ note: what now follows is the decision letter after the authors submitted for further consideration.]

*Major points:*

*1) Electrophysiology is performed in ACC, and the authors are able to distinguish NS, LS, and HS conditions based on firing rates using an SVM classifier. They also show that manipulation of ACC can influence some of their behavioral measures of pain responses. My main issue with these results is that there are likely many behavioral variables that correlate with pain and that influence the behavioral readouts. Just because pain is what is considered here, it does not mean that is what ACC is encoding or influencing. Although there are many possible correlated variables, I will give an example using a single one: movement / motor efference copy. When the laser is applied to the forelimb, this will result in forelimb movement (withdrawal or smaller movements) and these movements might vary substantially between NS, LS, and HS conditions. It is well established that much of cortex (even sensory areas) receives motor efference copies. It therefore seems entirely possible that the spiking activity measured in Figure 2 could be related to movement and not pain. The authors have not done any controls to rule out movement. They would need to show that movement is the same between conditions or would need to show that movements outside a pain context do not trigger ACC activity.*

We appreciate this comment from the reviewer. Our experiments are performed on freely moving animals, and our recordings have two phases – a baseline phase and a phase after noxious stimulation. Since we are analyzing changes in firing rates from baseline after peripheral stimulation, baseline locomotion is unlikely to affect our data interpretation. As the reviewer pointed out, however, pain does elicit additional movements. In our case, acute pain-induced movements are in the form of paw withdrawals. It should be noted that these paw withdrawals are well-known spinal reflexes (Negus et al., J Pharmacol Exp Ther. 2006; Vardeh et al., 2016), and hence are not movements directly produced by the motor system in the brain. As a result, compared with purposeful movements, spinal withdrawal reflexes are less likely to be accompanied by a motor efference copy in the brain. Nevertheless, as the reviewer inferred, these movements can potentially confound the interpretation of our neural findings, and we agree that control experiments are necessary to rule out such confounds.

To ensure that paw withdrawals did not influence our neural recordings, we performed the following control experiments. First, as the reviewer suggested, we analyzed the motor function in response to peripheral stimuli. We found that the percentages of withdrawal responses to LS and HS were both 100% (in contrast, the percentage of withdrawal responses after NS was <5%). We reported this data in the revised Results section. Next, as the reviewer suggested, we compared the motor aspect of paw withdrawals in response to LS and HS. To quantify this motor response, we calculated the velocity of paw withdrawals after laser stimulation using a high speed camera. We measured the paw withdrawal velocity by dividing the highest point each paw reached by the time it took to reach this point (see Materials and methods). We did not find any statistical difference in the withdrawal velocity between the responses to LS and HS, and this result was reported in Figure 2—figure supplement 2. This experiment suggests that there is no significant difference in the gross motor response to LS and HS, in contrast to the dramatic difference in neural spiking rates in the ACC seen in Figure 2.

To provide further support that ACC activities are specific to pain rather than movement, we also examined the effect of bidirectional modulation of ACC on movements, as the reviewer suggested. In Figure 3, we have shown that activation of the ACC neurons does not alter the latency to paw withdrawals, suggesting that ACC activation is unlikely to disrupt motor activities during our experiments. As suggested by the reviewer, we have done an additional experiment to measure locomotion over 10 minutes for rats that received optogenetic activation or inhibition of the ACC. The optogenetic protocol we used in this control experiment mirrors the ones used for CPA tests and neurophysiological recordings. Thus, light was turned on for 3 seconds every 10 seconds during the locomotion tests. We did not observe any difference in locomotion when the ACC was activated or inhibited. These results are reported in Figure 3—figure supplement 2 and Figure 4—figure supplement 2. The results of these control experiments suggest that ACC modulations do not alter locomotion.

In the context of these control experiments, the simplest interpretation of our neurophysiological data is that neural activities in the ACC correlated with nociceptive information, and not pain-induced movements. This interpretation is also compatible with findings from previous fMRI and animal studies that suggest a crucial role for the ACC in decoding the aversive component of pain.

At the same time, however, we acknowledge that we cannot absolutely rule out all the potential behavioral covariance associated with noxious stimulation, as the reviewer has suggested. As a result, we felt that it would be prudent to limit our interpretation and to avoid the broader claim for the identification of “pain-tuning neurons.” We have thus removed this term throughout our revised manuscript. In addition, we have put forth a statement limiting the breadth of interpretation of our neural data in the Discussion section of our revised manuscript.

*Relatedly, it is possible that activating or inactivating ACC causes changes in locomotor behavior. ACC is interconnected with motor regions and thus it might be possible that movement (or many other behavioral variables) could be perturbed rather than pain coding. Have the authors measured any features of locomotion in their experiments? Without at least ruling out movement cases, I am not convinced it is fair to conclude that ACC is encoding features directly related to pain. ACC could very well be encoding a different variable that just happens to covary with pain here.*

We appreciate this comment from the reviewer. In Figure 3, we have shown that activation of the ACC neurons does not alter the latency to paw withdrawals, suggesting that ACC activation is unlikely to disrupt locomotor activities. However, we agree with the reviewer that an additional locomotion control experiment is necessary. As suggested by the reviewer, we have measured locomotion over 10 minutes for rats that received optogenetic activation or inhibition of the ACC. The optogenetic protocol we used in this control experiment mirrors the ones used for CPA experiments. Thus, light was turned on for 3 seconds every 10 seconds during the locomotion tests. We did not observe any differences in locomotion when the ACC was activated or inhibited. These results are reported in Figure 3—figure supplement 2 and Figure 4—figure supplement 2. The results of these control experiments suggest that ACC modulations do not alter locomotion. Hence, they provide additional support for our interpretation that the ACC likely encodes features directly related to pain rather than movements.

*2) The optogenetic experiments seem to need additional controls. First, there are no measurements of what the ChR2 and NpHR stimuli do to neural activity. It is assumed that they activate and inactive ACC, respectively. However, it seems important to show evidence that this is the case. It is dangerous to assume this just because of behavioral effects. For example, it is well established with microstimulation that inhibition can be rapidly recruited through synaptic connections and actually shut down excitatory activity. It seems essential to have some validation of the tools.*

We would like to thank the reviewer for the suggestion for these controls. We have performed these control experiments. Specifically, we have performed in vivo optrode recordings with both ChR2 and NpHR. The results are shown in revised Figure 3—figure supplement 1 and Figure 4—figure supplement 1. As shown in these figures, ChR2 stimulation produces faithful neural activation, whereas NpHR stimulation results in depressed neuronal spiking.

*Also, it is common to do control experiments with laser light and a virus lacking the opsin. This controls for potential effects, like heating the brain or the visible light from the laser. For example, in Figure 3, the mice could be learning a paired association between the blue light (which is easy for them to see compared to the yellow light in Figure 4) and the pain from the laser to the forelimb (like in traditional fear conditioning). This pairing could drive their behavior in a more robust way than just the forelimb stimulus. Together these experiments seem important to verify that the effects are due to bidirectional modulation of ACC firing and not things like seeing the blue light.*

As suggested by the reviewer, we have performed a control experiment by injecting a viral vector that expressed only YFP without any opsins into the ACC. We applied laser light to the ACC in rats injected with this virus during the CPA tests. We paired one chamber with light treatment and LS, and the opposite chamber with LS alone. We also repeated this test with HS. We did not observe any preference or aversion for the chamber paired with light treatment. The results from these control experiments are reported in Figure 3—figure supplement 3. These results provide support for the interpretation that behavioral findings from Figure 3,Figure 4 are due to bilateral modulations of ACC firing as the result of functional activation of the opsins.

*3) The authors have quantified latency to withdrawal for the LS and HS stimuli. Have they also looked at the fraction of trials with a withdrawal?*

We have calculated the percentage of paw withdrawals in response to LS and HS. We found that the percentage of withdrawal responses in both cases were 100%. In contrast, the percentage of withdrawal responses after NS was <5%. We reported this data in the Results section.

*4) In all experiments, how was the laser stimulus calibrated for the LS and HS stimuli? Was the latency to withdrawal for LS and HS similar for all experiments in Figure 1–Figure 4?*

We calibrated our laser with a power meter prior to each behavior test or electrophysiological recording. LS stimulation corresponded to a power of 150mW, whereas HS stimulation corresponded to a power of 250mW from our laser. The latency to withdrawal for LS and HS is similar for all experiments. We have revised our manuscript to include this information in the Materials and methods section.

5) There are many places where statistics are missing. Statements are made about differences between figure panels but no statistics are provided. Some cases include:

*– Comparing Figure 1 in Results paragraph 3*

For this comparison, we did not provide statistics because we wanted to show qualitatively that there appears to be a difference in the avoidance of the LS paired chamber (compared with NS) between CFA-treated rats and rats without chronic pain. We then quantified this phenotypic difference by comparing saline- with CFA-treated rats in Figure 1. Thus, Figure 1 provides a statistical analysis for the difference suggested by Figure 1.

*– Comparing Figure 1 in the same section*

Please see the reply above. Figure 1 provides the statistical analysis for the qualitative difference suggested by Figure 1.

*– Comparing Figure 2, in subsection “Chronic pain disrupts the ACC representation of acute pain signals, paragraph two”*

We have omitted this comparison, as it was not central to our study. We have changed the Results section accordingly.

*– Comparing Figure 1,Figure 2, in subsection “Chronic pain impairs the bidirectional regulation of acute pain by the ACC”*

For this comparison, we did not provide statistics because we wanted to show qualitatively that there appears to be an increase in the avoidance of the LS paired chamber (compared with NS) in rats that received optogenetic treatment in the ACC. We then quantified this increased avoidance in Figure 3 by a calculation of the aversion score, and compared this aversion score with the aversion score for rats that experienced chronic pain (CFA-treated rats).

*– Comparing Figure 1,Figure 4, same section, paragraph three*

We have provided statistics for this analysis in the revised manuscript.

*– Comparing Figure 1,Figure 4, same section, paragraph three*

We have provided statistics for this analysis in the revised manuscript.

In some of the above referenced instances, we tried to demonstrate *qualitatively* the difference in pain aversion between two different conditions (either in rats that experienced chronic pain vs no chronic pain or in rats that received optogenetic modulation of the ACC vs rats that did not receive such modulation). In a sense, we called the reader’s attention to such qualitative differences, and we then performed more rigorous statistical analyses using appropriate controls. To avoid any confusion, we have clarified our approach in the Results section of the revised manuscript to indicate the distinction between qualitative comparison and quantitative analyses in the referenced sections of our revised manuscript.

*6) The authors mention a change in slope between the plots in Figure 2. In the legend, the slopes are noted, but no statistics are provided to test if these slopes are significantly different.*

We apologize for this oversight. We have reported the mean/SE estimate of the slope parameter in the revised manuscript. We have also provided the statistical analysis in the legend to Figure 2 as well as in the Materials and methods section.

*7) I am not convinced by the occlusion results from Figure 3. Both CFA and ChR2 on their own cause a higher CPA score. This is due to a decrease in time spent in the conditioned chamber. With both of these cases, the time spent in the conditioned chamber approaches zero. When CFA and ChR2 are done together, there is no chance of ever seeing an additive effect because each one individually already approaches the floor (zero time in the conditioned chamber). Given that there is no chance of seeing an additive effect due to floor effects, this result is not meaningful. I suggest removing Figure 3.*

We have now removed Figure 3 and the corresponding text in the revised manuscript.

*8) For the SVM analyses, it would be good to show a chance level of decoding. For example, if the labels for the NS and HS trials are randomized in Figure 2 (for example with 1000 runs of different random assignments of labels), what are the bounds of the chance level of decoding achieved? Do these values fall outside the chance levels?*

We appreciate this comment from the reviewer. We have now computed the chance level based on randomly permutated labels. The procedure is described in the revised Materials and methods section. One representative example is shown in Figure 2—figure supplement 3. The mean and the error bar of the chance level are shown in the figure. The decoding accuracy derived from the true labels is significantly greater than the chance levels in all cases.